

# Structural uncertainty in air mass factor calculation for NO₂ and HCHO satellite retrievals

Alba Lorente[1], K. Folkert Boersma[1,2], Huan Yu[3], Steffen Dörner[4], Andreas Hilboll[5,6], Andreas Richter[5], Mengyao Liu[7], Lok N. Lamsal[8], Michael Barkley[9], Isabelle De Smedt[3], Michel Van Roozendael[3], Yang Wang[4], Thomas Wagner[4], Steffen Beirle[4], Jin Tai Lin[7], Nickolay Kroktov[8], Piet Stammes[2], Ping Wang[2], Henk. J. Eskes[2], and Maarten Krol[1,10,11]

[1]Wageningen University, Meteorology and Air Quality Group, Wageningen, the Netherlands.
[2]Royal Netherlands Meteorological Institute, De Bilt, the Netherlands.
[3]Belgian Institute for Space Aeronomy (BIRA-IASB), Brussels, Belgium.
[4]Max-Planck Institute for Chemistry (MPI-C), Mainz, Germany.
[5]Institute of Environmental Physics (IUP-UB), University of Bremen, Bremen, Germany.
[6]MARUM-Center for Marine Environmental Sciences, University of Bremen, Bremen, Germany.
[7]Laboratory for Climate and Ocean-Atmosphere Studies, Department of Atmospheric and Oceanic Sciences, School of Physics, Peking University, Beijing 100871, China.
[8]Atmospheric Chemistry and Dynamics Laboratory, NASA Goddard Space Flight Center, Greenbelt, Maryland, USA.
[9]EOS Group, Department of Physics and Astronomy, University of Leicester, Leicester, UK.
[10]Netherlands Institute for Space Research (SRON), Utrecht, the Netherlands
[11]Institute for Marine and Atmospheric Research Utrecht, Utrecht University, Utrecht, the Netherlands

*Correspondence to:* A. Lorente
alba.lorentedelgado@wur.nl

**Abstract.** Air mass factor (AMF) calculation is the largest source of uncertainty in NO₂ and HCHO satellite retrievals in situations with enhanced trace gas concentrations in the lower troposphere. Structural uncertainty arises when different retrieval methodologies are applied in the scientific community to the same satellite observations. Here, we address the issue of AMF structural uncertainty via a detailed comparison of AMF calculation methods that are structurally different between seven

5   retrieval groups for measurements from the Ozone Monitoring Instrument (OMI). We estimate the escalation of structural uncertainty in every sub-step of the AMF calculation process. This goes beyond the algorithm uncertainty estimates provided in state-of-the-art retrievals, which address the theoretical propagation of uncertainties for one particular retrieval algorithm only. We find that top-of-atmosphere reflectances simulated by four radiative transfer models (RTMs) (DAK, McArtim, SCIATRAN and VLIDORT) agree within 1.5%. We find that different retrieval groups agree well in the calculations of altitude resolved

10   AMFs from different RTMs (to within 3%), and in the tropospheric AMFs (to within 6%) as long as identical ancillary data (surface albedo, terrain height, cloud parameters and trace gas profile) and cloud and aerosol correction procedures are being used. Structural uncertainty increases sharply when retrieval groups use their preference for ancillary data, cloud and aerosol correction. On average, we estimate the AMF structural uncertainty to be 42% over polluted regions and 31% over unpolluted regions, mostly driven by substantial differences in the a priori trace gas profiles, surface albedo and cloud parameters.

15   Sensitivity studies for one particular algorithm indicate that different cloud correction approaches result in substantial AMF differences in polluted situations (5 to 40% depending on cloud fraction and cloud pressure, and 11% on average) even for





low cloud fractions ( $< 0.2$ ) and the choice of aerosol correction introduces an average uncertainty of 50% for situations with high pollution and high aerosol loading. Our work shows that structural uncertainty in AMF calculations is significant and that is mainly caused by the assumptions and choices made to represent the state of the atmosphere. To point out which approach and which ancillary data are the best for AMF calculations, we call for well-designed validation exercises focusing on polluted situations when AMF structural uncertainty has the highest impact on $NO_2$ and HCHO retrievals.

# 1 Introduction

Satellite observations in the UV and visible spectral range are widely used to monitor trace gases such as nitrogen dioxide ($NO_2$) and formaldehyde (HCHO). These gases are relevant for air quality and climate change, because they are involved in the formation of tropospheric ozone and aerosols, which have an important influence on atmospheric radiative forcing (IPCC, 2013). Ozone and aerosols are defined as "essential climate variables" (ECVs) by the Global Climate Observing System (GCOS). These ECVs and their precursors ($NO_2$ and HCHO among others) are included in the ECV framework because they contribute to characterize Earth's climate and they can be monitored from existing observation systems (Bojinski et al., 2014). Currently a wide range of ECV products are available, but they rarely have reliable and fully traceable quality information. To address this need, the Quality Assurance for Essential Climate Variables project (QA4ECV, www.qa4ecv.eu) aims to harmonize, improve and assure the quality of retrieval methods for the ECV precursors $NO_2$ and HCHO. Here, we focus on retrievals of tropospheric $NO_2$ and HCHO vertical column densities (VCDs) from spaceborne UV/Vis spectrometers. Retrievals from these instruments have been used for a wide range of applications. These notably include estimating anthropogenic emissions of $NO_x$ and HCHO (e.g. Boersma et al. (2015), Marbach et al. (2009)), natural isoprene emissions (e.g. Marais et al. (2014), Barkley et al. (2013)) and $NO_x$ production from lightning (e.g. Lin (2012), Beirle et al. (2010)), data assimilation (e.g. Miyazaki et al. (2012)), and trend detection (e.g. Richter et al. (2005), De Smedt et al. (2010)).

Although trace gas satellite retrievals have improved over the last decades (e.g. Richter et al. (2011), De Smedt et al. (2012), Bucsela et al. (2013)), there is still a need for a more complete understanding of the uncertainties involved in each retrieval step. The retrieval of $NO_2$ and HCHO columns consists of three successive steps. First a spectral fitting is performed to obtain the trace gas concentration integrated along the atmospheric light path (slant column density, SCD) from backscattered radiance spectra. For $NO_2$, the stratospheric contribution to the SCD is separated to obtain the tropospheric SCD. Finally, the SCD is converted into the vertical column density (VCD) using an air mass factor (AMF). Previous studies indicated that the AMF calculation is the largest source of uncertainty (20-50% from typical VCDs uncertainties of 40-60%) in the $NO_2$ and HCHO retrievals in scenarios with a substantial tropospheric contribution to the total column (e.g. Boersma et al. (2004), De Smedt et al. (2008), Barkley et al. (2012)). These studies arrived at such theoretical uncertainty estimates based on error propagation for one specific retrieval algorithm.

Theoretical uncertainty (also known as parametric uncertainty) is the uncertainty arising within one particular retrieval method. Structural uncertainty is the uncertainty that arises when different retrieval methodologies are applied to the same data (Thorne et al., 2005). To represent the state of the atmosphere, several choices and assumptions are made in the retrieval



algorithm, in particular within the AMF calculation. Even though these choices are physically robust and valid, when different retrieval algorithms based on different choices are applied to the same satellite observations, this usually leads to different results. The structural uncertainty is intrinsic to the retrieval algorithm formulation and it is considered to be a source of systematic uncertainty (Povey and Grainger, 2015). In principle, theoretical and structural uncertainties should be considered independently from each other. However, in the calculation of the theoretical uncertainty, the contribution of the ancillary data

is often calculated comparing different databases (e.g. to estimate surface albedo uncertainty as in Boersma et al. (2004)) rather than using the uncertainty of the database itself. Consequently, some components are shared in the structural and theoretical uncertainty calculations. However, for a full structural uncertainty estimate, all sources of methodological differences need to be considered. In the framework of AMF calculations addressed here, this implies e.g. the selection of radiative transfer model, vertical discretization and interpolation schemes, the method for cloud and aerosol correction and the selection of (external

or ancillary) data on the atmospheric state (surface reflectivity, cloud cover, terrain height, and a priori trace gas profile). The problem of structural uncertainty has been addressed in other fields of atmospheric sciences, e.g. in satellite retrievals for atmospheric variables (Fangohr and Kent, 2012) and in numerical models for climate studies (Tebaldi and Knutti, 2007).

There are few studies addressing structural uncertainty for trace gas retrievals. Van Noije et al. (2006) compared $NO_2$ tropospheric columns retrieved from GOME data by 3 different groups. In that study, the discrepancies inherent to differences and

assumptions in the retrieval methods were identified as a major source of systematic uncertainty. However, the causes of discrepancies between retrievals were not addressed but were targeted for a more detailed investigation. In this study we focus on AMF structural uncertainty, by comparing the AMF calculation approaches by seven different retrieval groups and providing a traceable analysis of all components of the AMF calculation. Ensemble techniques to estimate structural uncertainty have already been applied in different atmospheric disciplines (e.g. Steiner et al. (2013), Liu et al. (2015)). The groups that partici-

pated in this study are: Belgian Institute for Space Aeronomy (IASB-BIRA; abbreviated as BIRA), Institute of Environmental Physics, University of Bremen (IUP-UB), Wageningen University (WUR) and Royal Netherlands Meteorological Institute (KNMI) (calculations made by WUR following the KNMI approach, abbreviated as WUR), University of Leicester (UoL), Max Planck Institute for Chemistry (MPI-C), NASA Goddard Space Flight Center (NASA-GSFC; abbreviated as NASA) and Peking University.

We start with a comparison of top-of-atmosphere (TOA) reflectances simulated by radiative transfer models (RTMs), the main tool for any AMF calculation (Sect. 3.1). The RTMs DAK, McArtim, SCIATRAN and VLIDORT solve the radiative transfer equation differently, and have different degrees of sophistication to account for Earth's sphericity and multiple scattering. Next we compare altitude-dependent (or box-) AMFs for $NO_2$ and HCHO computed with the four RTMs (Sect. 3.2). This is followed by a comparison of tropospheric AMFs (for $NO_2$) calculated by four groups for measurements by the Ozone

Monitoring Instrument (OMI) based on identical settings (same ancillary data and same approach for cloud and temperature correction) (Sect. 3.3.1). We interpret the resulting spread between the tropospheric AMFs as the AMF structural uncertainty associated with using different RTMs, vertical discretization and interpolation schemes. Then, we investigate how the choice of cloud correction affects the AMF structural uncertainty (Sect. 3.3.2). For the overall structural uncertainty estimate, we perform a round robin exercise (Sect. 3.3.3), in which seven different groups calculate $NO_2$ AMFs using their own preferred



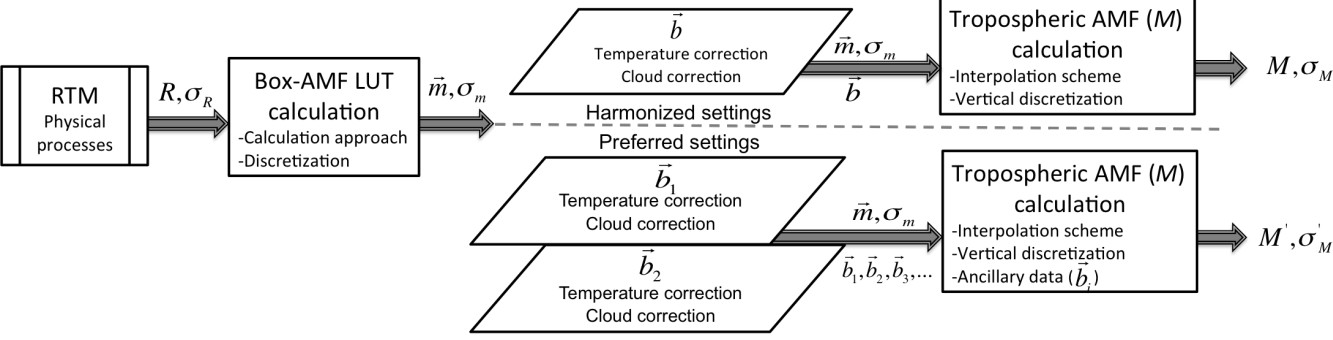

**Figure 1.** Flowchart of AMF calculation and comparison process followed in the study. In the third step forward model parameters (**b**: surface albedo, surface pressure, a priori profile, temperature, cloud fraction and cloud pressure) are selected for harmonized settings comparison (upper part) and preferred settings comparison (lower part). In each step the main differences between the compared elements are highlighted. The compared parameters and their structural uncertainty ($\sigma$) in each step are: TOA reflectance ($R$, $\sigma_R$), box-AMFs ($\boldsymbol{m}$, $\sigma_m$), and tropospheric AMFs ($M$, $\sigma_M$).

methods for cloud and aerosol correction and sources of ancillary data. Here we asses the effect of the different choices in the AMF structural uncertainty. Finally, we investigate how stratospheric AMFs are affected by the selection of RTM and their physical description of photon transport through a spherical atmosphere. The complete chain of uncertainties associated with each phase provides traceable quality assurance for the AMF calculation. Recommendations on best practices are given for this particular algorithm step and they will be applied in a community best practice retrieval algorithm for ECV precursors, under

development in the framework of the QA4ECV project.

## 2   Methods

### 2.1   AMF calculation process

The concept of traceability chain (here in the form of a flow diagram) for the AMF calculation process and uncertainty assessment used in this study is illustrated in Fig. 1. Structural uncertainty estimated in each step are based on the standard deviation

($1\sigma$) of relative differences of the compared elements. Modelled reflectance ($R$) at TOA is the starting point for air mass factor calculations using radiative transfer models. A RTM solves the radiative transfer equation, which describes the transport of radiation through the atmosphere to the observer (in our case the satellite) and the physical processes that affect the intensity of the radiation (absorption, scattering, refraction and reflection) (first box in the diagram in Fig.1). Reflectance (unitless) is calculated from fundamental radiation quantities, and it is defined as the ratio of modelled Earth radiance ($I$) (times $\pi$) and the



solar irradiance at TOA perpendicular to the solar beam ($E_0$) multiplied by the cosine of the solar zenith angle ($\mu_0$):

$$R(\lambda) = \frac{\pi I(\lambda)}{\mu_0 E_0(\lambda)} \tag{1}$$

Different models use different methods to solve the radiative transfer equation and to describe the sphericity of the Earth's atmosphere. Differences in modelled TOA reflectances between RTMs provide an estimate for the reflectance structural uncertainty ($\sigma_R$). This uncertainty due to the choice of the RTM propagates to the next step in the AMF calculation.

Altitude dependent AMFs (box-AMFs) characterize the vertical sensitivity of the measurement to a certain trace gas (e.g., Palmer et al. (2001)). They are directly related to how the measured radiance at TOA changes with a change of the optical depth of the atmosphere (related to the presence of a trace gas in a certain atmospheric layer), with the requirement that the absorber is optically thin (optical thickness $\tau_{\text{gas}} \ll 1$). In the context of the AMF calculation (second box in diagram of Fig. 1), box-AMFs for each layer can be calculated and stored in a look-up table (LUT) as a function of the forward model parameters (**b**) such as satellite viewing geometry, pressure level, surface pressure and surface reflectivity. There is also the possibility of online radiative transfer calculations for determining box-AMFs, i.e., bypassing the calculation of a LUT (e.g., Lin et al., (2014, 2015); Hewson et al. (2015)). Different RTMs use different vertical discretizations of the atmosphere, and calculate box-AMFs in different ways (see Sect. 2.2). A comparison of the box-AMF LUTs calculated with different RTMs provides a measure for the box-AMF structural uncertainty ($\sigma_m$), which can be considered as the reproducibility of the box-AMFs from different RTMs when the same settings and input data are used.

The air mass factor ($M$) represents the length of the mean light path at a certain wavelength for photons interacting with a certain absorber in the atmosphere relative to the vertical path. AMFs are used to convert the SCD obtained from the reflectance spectra to a VCD. To calculate the tropospheric VCD, a tropospheric AMF is used ($\text{VCD}_{tr} = \text{SCD}_{tr}/M_{tr}$). But for species that have a stratospheric contribution to the total slant column, the stratospheric SCD first needs to be estimated and substracted from the total SCD. For this purpose, a stratospheric AMF is often used together with an independent estimate of the stratospheric VCD (e.g. from a chemistry transport model, a climatology or independent measurements) ($\text{SCD}_{strat} = \text{VCD}_{strat} \cdot M_{strat}$).

If the trace gas is optically thin, the total air mass factor can be written as the sum of the box-AMFs of each layer weighted by the partial vertical column (e.g., Palmer et al. (2001), Boersma et al. (2004)):

$$M = \frac{\sum_l m_l(\widehat{\mathbf{b}}) x_{a,l}}{\sum_l x_{a,l}} \tag{2}$$

In Eq. 2 $m_l$ is the box-AMF and $x_{a,l}$ is the trace gas sub-column in layer $l$. However, as the actual profile of sub-columns is unknown, an a priori profile has to be used in the AMF calculation. The summation is done over the atmospheric layers ($l$) of the a priori trace gas profile. In this step of the AMF calculation, apart from the profile shape of the trace gas, it is also necessary to have the best estimates for other forward model parameters ($\widehat{\mathbf{b}}$) such as satellite viewing geometry, surface pressure and surface reflectivity. Surface reflectivity depends on the surface properties and the geometry of the incident and reflected light. This anisotropy is described by the bidirectional reflectance distribution function (BRDF). In practice, surface reflectivity is





often approximated by an isotropic Lambertian equivalent reflector (LER). There are different sources from which the a priori information can be obtained. It is desirable to use as much information as possible retrieved from the satellite instrument itself. This practice gives consistency to the trace gas retrieval regarding the forward model parameters.

The $NO_2$ and HCHO absorption cross sections used in the SCD fit and box-AMF calculation are representative for one fixed temperature. However, these cross sections vary with temperature, so it is necessary to apply a temperature correction. This correction accounts for the change in the absorption cross section spectrum as a function of the effective temperature at a specific layer, based on temperature and trace gas profiles from model data or climatologies (see Eq. S1 in the supplement). The correction is commonly done by applying a correction factor ($c_l$) for each layer in the AMF calculation.

$$M = \frac{\sum_l m_l(\widehat{\mathbf{b}}) x_{a,l} \cdot c_l}{\sum_l x_{a,l}} \tag{3}$$

Most of the studies in which the temperature effect on the $NO_2$ cross section is analyzed assume a simple dependency of the correction factor to temperature (Vandaele et al., 2002) (see Eq. S2 and S3 for typically used correction factors). For satellite applications, the change of the absorption cross section in case of $NO_2$ has been reported to be approximately -0.3% per K in the visible (Bucsela et al. (2013), Boersma et al. (2002)) and -0.05% per K for HCHO (De Smedt, 2011).

Satellite retrievals also need to consider the presence of clouds. In the AMF calculation, residual clouds can be accounted for via the independent pixel approximation (IPA) or via cloud masking (CM). The IPA consists of calculating the AMF for a partly cloudy scene as a linear combination of cloudy ($M_{cl}$) and clear ($M_{cr}$) components of the AMF, weighted by the cloud radiance fraction $w$ (i.e. the fraction of radiance that originates from the cloudy part of the pixel) (Martin et al. (2002), Boersma et al. (2004)):

$$M = wM_{cl} + (1-w)M_{cr} \tag{4}$$

In Eq. 4 $w$ is wavelength dependent through radiation intensity, so it will be different for $NO_2$ and HCHO (see Eq. S4 in the supplement). Here, AMFs for cloudy scenes are calculated using Eq. 3 with a specific cloud albedo and cloud pressure, with $m_l = 0$ below the cloud. In line with assumptions made in current cloud retrievals, the cloud is considered as a Lambertian reflector with a fixed cloud albedo. This simple cloud model is in most cases suitable to be used in trace gas retrieval algorithms (Acarreta et al., 2004). In the cloud masking method, the atmosphere is assumed to be cloud-free for cloud radiance fractions below a certain threshold (e.g. 0.5, Richter et al. (2011)) and measurements with larger cloud fraction are discarded. In that case, Eq. 4 reduces to $M = M_{cr}$. In both approaches accurate information is needed on the cloud radiance fraction and in the IPA approach also on the effective cloud pressure.

Different retrieval groups use different sources for the ancillary data, as well as different methods to account for the temperature dependence and the presence of clouds and aerosols (e.g. Van Noije et al. (2006)). In our study, each of the groups first calculated tropospheric AMFs using harmonized settings, i.e. using the same forward model parameters, temperature correction and cloud correction. In order to calculate the total AMF using Eq. 3, an interpolation from the LUT needs to be done to obtain the box-AMFs at the specific values of the forward model parameters. Furthermore, a vertical interpolation is required to



**Table 1.** Overview of radiative transfer models that participated in the top-of-atmosphere reflectance comparison and their main characteristics.

| Model | DAK | McArtim | SCIATRAN | VLIDORT |
|---|---|---|---|---|
| Reference | Stammes (2001) | Deutschmann et al. (2011) | Rozanov et al. (2014) | Spurr et al. (2001) |
| Institute | KNMI, WUR | MPI-C | IUP-UB | IASB-BIRA |
| Solving the Radiative Transfer equation | Doubling adding method | Monte Carlo methods to solve integral form of RTE | Source function integration technique and discrete - ordinate method | Linearized discrete ordinate solution |
| Sphericity correction | Pseudo spherical for direct solar incident photons | Full 3D spherical model calculations on a sphere | Full spherical mode for solar and single scattered photons | Pseudo spherical for solar and single scattered photons |

adjust the vertical discretization of the a priori absorber profile to the one of the LUT. From the comparison of the tropospheric AMFs calculated using harmonized settings, we can thus obtain a relative AMF structural uncertainty, which is determined by
different approaches in interpolation and vertical discretization of the box-AMFs, assuming that the selected forward model parameters are the true values.

Next, each of the groups used their preferred settings to calculate tropospheric AMFs. In this round-robin exercise, a comparison of state-of-the-art retrieval algorithms, the differences between AMFs not only arise from differences between the RTMs, vertical discretization and interpolation but also from differences in the selection of forward model parameter values and the
different corrections for clouds, aerosols and surface reflectivity. Thus the differences in the AMFs using preferred settings can be interpreted as the overall structural uncertainty of the AMF calculation (Thorne et al., 2005).

## 2.2 Participating models

Four RTMs from different research groups participated in the comparison. Some differences between models are highlighted in Table 1. A brief summary for each model is listed alphabetically in this section and more detailed information about the
models can be found in the references.

**DAK**

DAK (Doubling-Adding KNMI) was developed at the Royal Netherlands Meteorological Institute (Stammes, 2001). DAK uses the doubling-adding method for solving the radiative transfer equation (Stammes et al. (1989), de Haan et al. (1987)). The method consists of first calculating the reflection and transmission properties of a homogeneous layer by repeated doubling,
starting with a very thin layer, and then adding homogeneous layers on top of each other, which then yields the reflection and transmission of the combined layers. The internal radiation field is computed at the interface of all layers and the radiation



emerging at the top of the atmosphere and at the surface is calculated. DAK accounts for multiple scattering and polarization. It is also possible to account for Earth's sphericity using the pseudo spherical option, which corrects for sphericity in the light path of the direct solar beam, but not in the scattered beam.

Box-AMFs are calculated with DAK in this study by WUR/KNMI by differencing the logarithm of reflectances at TOA with and without the trace gas in atmospheric layer $l$ divided by the gas absorption optical thickness of the layer $\tau_{gas}$:

$$m_l = -\frac{\ln R(\tau_{gas,l}) - \ln R(\tau_{gas,l} = 0)}{\tau_{gas,l}} \tag{5}$$

**McArtim**

McArtim (Monte Carlo Atmospheric Radiative Transfer Inversion Model) (Deutschmann et al., 2011) was developed at Univer-

sity of Heidelberg and Max-Planck Institute for Chemistry (MPI-C, Mainz). It is based on the backward Monte Carlo method: a photon emerges from a detector in an arbitrary line of-sight direction and is followed in the backward direction along the path until the photon leaves the top of the atmosphere. The various events which may happen to the photon at various altitudes are defined by suitable probability distributions. At each scattering event the probability that the photon is scattered into the direction of the Sun is calculated and the intensity of the photon is weighted by the sum of the probabilities of all scattering

events (local estimation method). In this RTM, the integro-differential equation for radiative transfer is deduced and solved using Neumann series, the summands of which are linked with the contributions of multiple scattering orders to the radiation field. McArtim is a 3D-model and uses full spherical geometry, which means that sphericity is accounted for incoming, single scattered and multiple scattered photons. The model is capable of including polarization and rotational Raman scattering (which are included in the simulations shown in this study).

Box-AMFs calculated by MPI-C are obtained from Jacobians (derived by $W = \frac{\partial \ln I}{\partial \beta}$, with $\beta$ (km$^{-1}$) the absorption coefficient) for each grid box according to the formula:

$$m_l = -\frac{W}{I \Delta h} \tag{6}$$

In Eq. 6 $W$ refers to the Jacobian (km), $I$ is the simulated radiance at TOA normalized by the solar spectrum (unitless) and $\Delta h$ is the grid box thickness (km).

**SCIATRAN**

SCIATRAN (Rozanov et al., 2014) was developed at the Institute of Environmental Physics at the University of Bremen (IUP-UB) in Germany. It models radiative transfer processes in the atmosphere from the UV to the thermal infrared, in both scalar and vector mode, i.e. with the option to account for polarization. The simulations can be done for a plane parallel, pseudo-spherical or fully spherical atmosphere. In the fully spherical approach, the integral radiative transfer equation is solved accounting for

single scattering in spherical mode, and multiple scattering is approximated with a solution of the differential-integral radiative transfer equation in the plane parallel mode.

SCIATRAN calculates the Jacobians or weighting functions, which are the derivatives of the simulated radiance with respect to atmospheric and surface parameters (air number density in this case). These quantities are related to the box-AMFs calculated





by IUP-UB as follows:

$$m_l = -\frac{W_l}{I\sigma\Delta h_l} \tag{7}$$

$W_l$ (W $\cdot$ m$^{-2}\cdot$nm$^{-2}\cdot$sr$^{-1}$/molec$\cdot$cm$^{-3}$) is the weighting function at atmospheric level $l$, $I$ (W $\cdot$ m$^{-2}\cdot$nm$^{-2}\cdot$sr$^{-1}$) is the TOA radiance, $\sigma_l$ (cm$^2$/molec) is the absorber cross section and $\Delta h_l$ (cm) is the thickness of the layer.

## VLIDORT

VLIDORT (Vector-LInearized Discrete Ordinate Radiative Transfer) was developed by Rob Spurr at RT SOLUTIONS, Inc. The model is based on the discrete ordinate approach to solve the radiative transfer equation in a multi-layered atmosphere, reducing the RTE to a set of coupled linear first order differential equations. Then, perturbation theory is applied to the discrete ordinate solution (Spurr et al., 2001). Intensity and partial derivatives of intensity with respect to atmospheric parameters and surface parameters (i.e. weighting functions) are determined for upwelling direction at TOA, for arbitrary angular direction. The pseudo spherical formulation in VLIDORT corrects for the curved atmosphere in the solar and scattered beam (for single scattering, not for multiple scattering).

Box-AMFs are derived from the altitude-dependent weighting functions determined by VLIDORT:

$$m_l = \frac{\partial \ln I}{\partial \tau_{gas,l}} = (\tau_{gas,l} \cdot \frac{\partial I}{\partial \tau_{gas,l}})/(I \cdot \tau_{gas,l}) \tag{8}$$

$I$ (W $\cdot$ m$^{-2}\cdot$nm$^{-2}\cdot$sr$^{-1}$) is the TOA radiance, $\tau_{gas}$ is the trace gas absorption optical thickness of the layer and the term $(\tau_{gas,l} \cdot \frac{\partial I}{\partial \tau_{gas,l}})$ is the altitude dependent weighting function.

## 3 Results

### 3.1 TOA reflectances

As a first exercise, a base case calculation and comparison of TOA reflectances was made to assess the performance of the four RTMs and to obtain the structural uncertainty in TOA reflectance modelling. The base case comparison allowed us to establish the best possible level of agreement between RTMs by identifying differences in the RTMs performance that in more complex settings would be difficult to recognize. Furthermore, total and ozone optical thickness were compared to evaluate how the models agreed in their treatment of scattering and absorption processes and whether differences in scattering and absorption can explain possible differences between the TOA reflectances.

Basic model parameters were established as input in all RTMs (details can be found in Table S1 in the supplementary material). The basic atmospheric profile was a 33-layer mid-latitude summer atmosphere (Anderson et al., 1986), and every group performed their own vertical discretization of this profile. In the RT modelling, we considered a clear sky atmosphere, so clouds and aerosols were not included. Rayleigh scattering and $O_3$ absorption were included, but Raman scattering was not included. The temperature dependence of the ozone cross-section was neglected in the reflectance calculation. TOA reflectances were




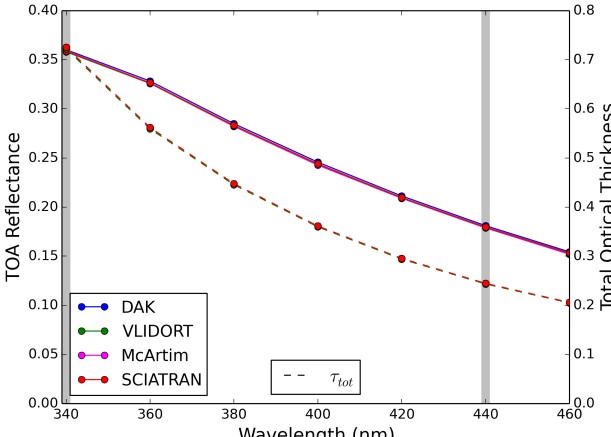

**Figure 2.** TOA reflectances simulated by 4 RTMs for $\theta_0 = 37°$ ($\mu_0 = 0.8$), off-nadir viewing angle $\theta = 72.5°$ ($\mu = 0.3$) and $\varphi = 0°$ as a function of wavelength (in 20 nm steps). Dashed lines represent total optical thickness computed by each RTM. Grey bands indicate the relevant wavelengths for HCHO (340 nm) and $NO_2$ (440 nm). Surface albedo is 0 and surface pressure is 1013 hPa.

calculated at 7 wavelengths, including 440 and 340 nm which are relevant for the retrievals of $NO_2$ and HCHO, respectively. Both scalar (i.e. without polarization) and vector (i.e. with polarization) calculations were performed in most of the cases. All

models applied their particular sphericity treatments to the calculations. The surface was considered as a Lambertian reflector by all the RTMs. This approximation assumes that surface reflectivity is isotropic (i.e it does not consider the directionality of the surface reflectance distribution). The selected geometries covered a wide range of values for solar zenith angle (SZA, $\theta_0$), viewing zenith angle (VZA, $\theta$), and relative azimuth angle (RAA, $\varphi = 180° - |\phi - \phi_0|$, where $\phi - \phi_0$ is the viewing direction minus solar direction). All the angles are specified with respect to the surface. The values for SZA span the typical range of

what UV/Vis sensors are encountering in orbit, and the maximum value of VZA is related to the higher possible values of this parameter for the future TROPOMI instrument (72.5°) (van Geffen et al., 2016).

All models calculate the same spectral dependency of TOA reflectance, as shown in Fig. 2 (solid line). TOA reflectance increases towards shorter wavelengths due to stronger Rayleigh scattering. TOA reflectance simulated by the different models agree within 1.3% for the geometries included in Fig. 2. The dashed line in Fig. 2 shows the total optical thickness as a function

of wavelength for DAK, SCIATRAN and VLIDORT (McArtim does not provide this output), and is generally consistent within 0.15% for all wavelengths except 340 nm, where the differences are 0.5%.

Figure 3 shows the distribution of relative differences (defined as (100(a-b)/a)) between TOA reflectances simulated by the four RTMs at 340 nm and 440 nm. The distribution is determined by the relative differences between all combinations of model differences, including all simulated geometry scenarios for a surface albedo of 0 and terrain pressure of 1013 hPa.

According to the standard deviation in both distributions (dashed lines in Fig. 3), the relative differences are below 1.5% at 340 nm and 1.1% at 440 nm in most geometry configurations (80% of the samples of the distribution), including the most common retrieval scenarios. The tails of the distributions at both wavelengths correspond to extreme viewing geometries, i.e.





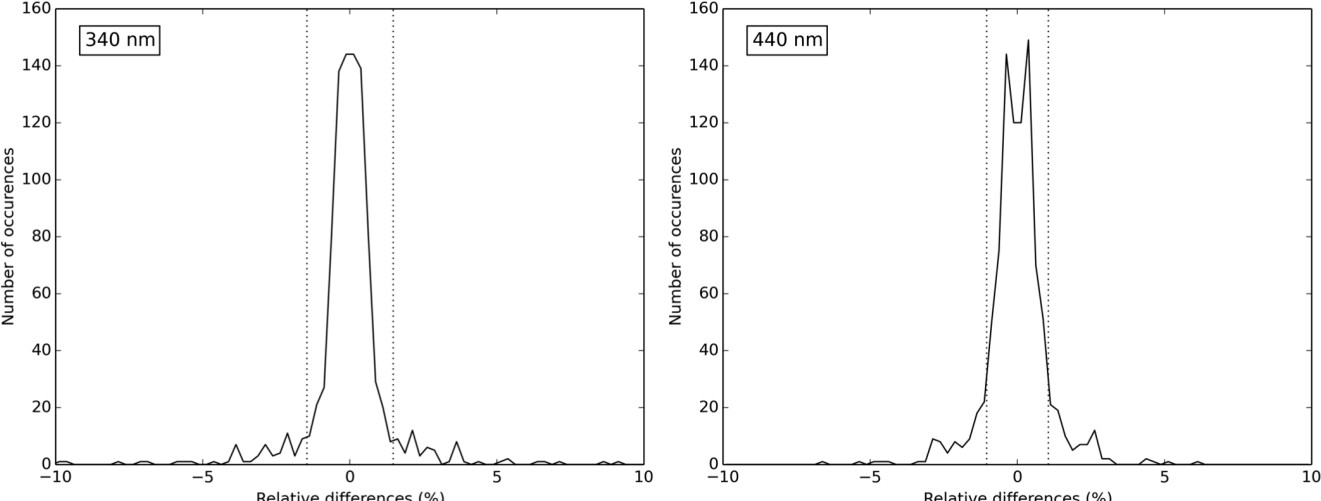

**Figure 3.** Distribution of relative model differences between TOA reflectances simulated by four RTMs including polarization (DAK-VLIDORT, DAK-SCIATRAN, DAK-McArtim, VLIDORT-SCIATRAN, VLIDORT-McArtim, SCIATRAN-VLIDORT and reversed combinations) for all geometry combinations ($0° < \theta_0 < 90°$, $\theta = 0°$, $72.5°$ and $\varphi = 0°$, $60°$, $90°$, $120°$, $180°$) (see Table 1 for exact values) at 340 nm (left panel) and 440 nm (right panel). The dashed lines represent the median plus/minus the standard deviation of the distribution. Surface albedo is 0 and surface pressure is 1013 hPa. Sample size in each distribution is 960.

for scenarios in which solar and viewing zenith angles are both large. Mean relative differences over all RTM pairs are at most 6.4% for extreme geometries ($\theta_0 = 87°, \theta = 72.5°$), and for shorter wavelengths. For nadir view ($\theta = 0°$) relative differences

are on average two times smaller than for larger VZA ($\theta \geq 60°$) at both 340 and 440 nm.

The results show strong consistency of TOA reflectance calculations for the most common moderate viewing geometry retrieval scenarios. Relative differences are somewhat higher for larger VZA, SZA and shorter wavelengths. For the more extreme geometries, the light path through the atmosphere is generally longer and photons have higher probability of undergoing interactions (scattering, absorption) with the atmosphere. Furthermore, differences in the treatment of Earth's sphericity for

the extreme geometries have a stronger influence than in close to nadir viewing geometries. These differences will still be present in the box-AMFs comparison in Sect. 3.2. Rayleigh scattering also affects the effective photon path and it is stronger at 340 nm than at 440 nm. Thus, small differences in the description of Rayleigh scattering in the RTMs are more likely to lead to differences for the extreme geometries and shorter wavelengths. The standard deviation of differences between modelled TOA reflectances of 1.5% (at 340 nm) and 1.1% (at 440 nm) in this comparison can be considered as the reflectance structural

uncertainty. The agreement in this study is better than in previous RTM comparisons like Wagner et al. (2007) and Stammes (2001) which reported differences of 5%. The detailed RTM comparison will serve as a test bed to analyze the performance of other RTMs.



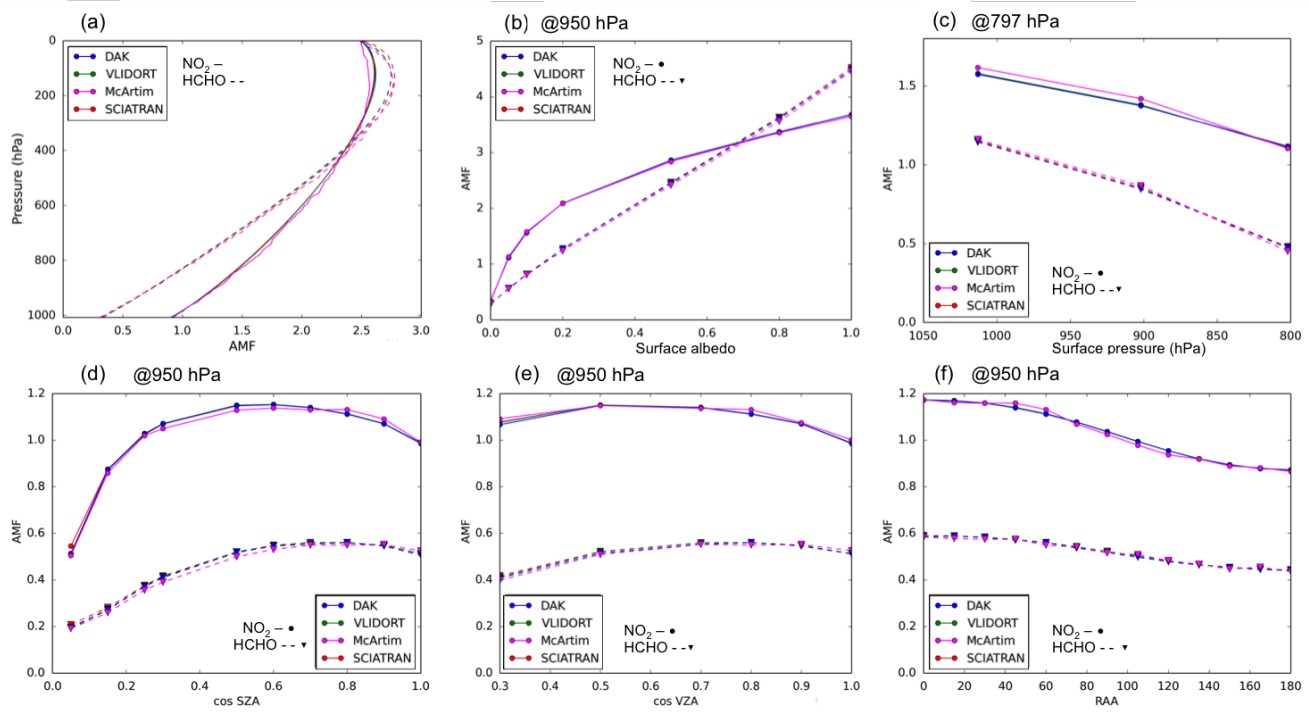

**Figure 4.** Box-AMF dependencies on forward model parameters for $NO_2$ at 440 nm (solid lines, circles) and HCHO at 338 nm (dashed lines, triangles) for a clear-sky atmosphere. (a) Box-AMFs vertical profile; (b) 950 hPa box-AMF as a function of surface albedo; (c) 797 hPa box-AMF as a function of surface pressure; (d) 950 hPa box-AMF as a function of cosine of SZA, (e) 950 hPa box-AMF as a function of cosine of VZA, (f) 950 hPa box-AMF as a function of RAA. In all panels the fixed parameters are: $\mu_0 = \mu = 0.8$ ($\theta_0 = \theta = 37°$), $\varphi = 60°$, surface albedo = 0.05, surface pressure = 1013 hPa.

### 3.2 $NO_2$ and HCHO altitude-dependent (box-) air mass factors

To calculate box-AMFs, a common vertical grid was agreed between the groups in order to reduce the sources that might cause

differences between the RTMs. The common profile resolution was 0.1 km from the surface up to 10 km, 1 km resolution from 10 to 60 km and 2 km resolution from 60 to 100 km. $NO_2$ box-AMFs were calculated at 440 nm and HCHO box-AMFs were calculated at 338, 341 and 344 nm to investigate the wavelength dependency (not shown). Box-AMFs were calculated accounting for polarization of light and Earth's sphericity. The number of reference points for surface albedo was increased and several surface pressures were added relative to the TOA reflectance simulations in the previous section to cover a wider

range of scenarios. All settings are detailed in Table S2.

Figure 4(a) shows that the 4 participating groups generally agree well on the vertical profile shape of $NO_2$ and HCHO box-AMFs in the troposphere. Measurement sensitivity decreases towards the surface, due to the increase of light scattering in the lower troposphere. Measurement sensitivity to HCHO is substantially lower than to $NO_2$, because of stronger Rayleigh





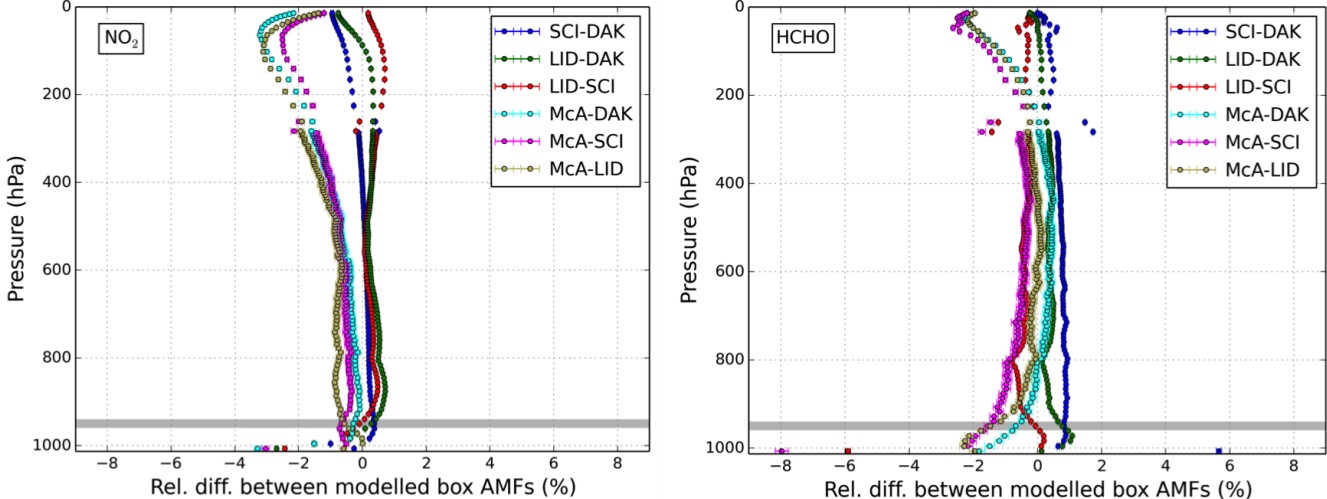

**Figure 5.** Vertical profile of mean relative differences between $NO_2$ box-AMFs (left) and HCHO box-AMFs (right) from DAK, McArtim, SCIATRAN and VLIDORT for a wide range of satellite viewing geometry ($0° < \theta_0 < 75°, 0° < \theta < 72.5°, 0° < \varphi < 180°$), surface albedo = 0.05 and surface height 1013 hPa. Grey bands indicate 950 hPa atmospheric layer.

scattering at shorter wavelengths. McArtim box-AMFs have lower values in the stratosphere (pink line), presumably reflecting

the more realistic description of atmosphere's sphericity in McArtim relative to the other models (see Sect. 3.4 for specific sphericity effect on AMFs). The vertical profile of McArtim shows a wavering line due to the statistical noise in the Monte Carlo simulations (which can be reduced by increasing the number of simulations). Figure 4 (b), (d)-(f) shows the $NO_2$ and HCHO box-AMF dependency on forward model parameters (surface albedo, surface pressure, SZA, VZA and RAA) in the lower troposphere at 950 hPa. This pressure level (close to the surface) is especially relevant because this is where trace gas

concentrations are enhanced in polluted situations. The sensitivity to surface albedo at 950 hPa (Fig. 4(b)) is similar for all four RTMs. Box-AMFs increase with surface albedo due to a stronger reflection of light at the surface. This increase is particularly strong for low values of surface albedo. For an albedo of 0.05, an increase of 0.01 in the surface albedo results in an increase of $11\%$ in the $NO_2$ box-AMF at 440 nm and of $9\%$ in the case of HCHO at 338 nm. The increase in the box-AMFs is less steep for higher values of surface albedo. Thus, an accurate knowledge of surface albedo is required especially for low albedo

values. For surface pressure (Fig. 4(c)), the box-AMF (at 797 hPa) decreases with decreasing surface pressure. For increasing terrain height, the amount of light scattered and reflected from below 797 hPa decreases. In a more elevated terrain, the photons undergo fewer scattering events, which tends to reduce box-AMFs at a specific level. Models agree well in representing this sensitivity. An error in the surface pressure of 10 hPa leads to $\pm 2\%$ errors in the lower tropospheric box-AMF values, which indicates the importance of accurate surface pressure information that is representative for the entire pixel area. Box-AMFs at

950 hPa show relatively weak dependency on VZA (Fig. 4(e)) and RAA (Fig. 4(f)) and stronger dependency on high values of SZA (Fig. 4(d)), but all RTMs agree well on measurement sensitivity to geometry parameters.





Figure 5 shows the vertical profile of mean relative differences in $NO_2$ (left panel) and HCHO (right panel) box-AMFs between all the models, for a specific surface albedo and surface height and a wide range of solar and viewing geometries. Generally, models reproduce box-AMFs to within 2% for $NO_2$ and 2.6% for HCHO. Mean relative differences are higher at

the lowest layers and around 300 hPa. This is due to unavoidable slight differences in vertical discretization of the surface-atmosphere boundary and where the resolution changes from 0.1 to 1 km at 10 km altitude in the different models. Specific differences were also found in the mid-upper troposphere and stratosphere, where McArtim is on average lower than the other RTMs. Those differences illustrate the different treatment of multiple scattering within the models. McArtim accounts for multiple scattering in a fully spherical atmosphere, whereas DAK, VLIDORT and SCIATRAN simulate multiple scattering

in a plane parallel atmosphere. In a spherical atmosphere, less light is horizontally scattered into the line of sight of the instrument than in a plane parallel atmosphere (see Fig. S2), which is one of the reasons for lower box-AMFs by McArtim in the stratosphere (visible in Fig. 4(a) between 200-0 hPa).

Relative differences for 950 hPa box-AMFs are below 1.1% for $NO_2$ and below 2.6% for HCHO in most geometry configurations (according to the standard deviation of relative differences distribution for 950 hPa box-AMFs, not shown). Higher

relative differences mainly occur between McArtim and the other models. The highest relative differences occur for scenarios with high VZAs ($\theta = 72.5°$) (not shown), again indicating that different Rayleigh scattering description and sphericity treatments in the radiative transfer modeling of the atmosphere are important.

This comparison indicates a good agreement between box-AMF LUTs computed using different RTMs. The structural uncertainty in the AMF calculation due to the choice of RTM and different interpolation schemes is 2% for $NO_2$ and 2.6%

for HCHO. These results suggest that a correct treatment of the processes affecting the effective light path in the atmosphere is important for box-AMF calculation. The vertical discretization is also relevant in box-AMF calculations, as demonstrated by the differences at specific altitudes (Fig. 5) and by the box-AMF sensitivity to altitude (Fig. 4 (a)). Therefore, the vertical sampling of the LUT should have a fine resolution, especially in the lower troposphere where strong gradients in $NO_2$ and HCHO concentrations occur. The dependencies of the box-AMFs at low surface albedo values (Fig. 4(b)) and to surface

pressure (Fig. 4 (c)), suggest that the number of reference points in the LUT for these parameters should be large.

### 3.3 Tropospheric air mass factors

In order to compute tropospheric AMFs via Eq. 3 we need to interpolate the box-AMFs from the LUT for the best estimate of the forward model parameters **b**. Generally a 6-D linear interpolation (or 5-D if the vertical resolution of the LUT and the a priori profile vertical grid are equal) is done over all the parameters on which the box-AMF depend. For each dimension, the

two closest values to the exact pixel parameters are used to obtain the interpolated box-AMF ($m_l$ in Eq. 3). This approach will introduce systematic errors in case of nonlinear dependencies of the parameters in the LUT. Pixel-by-pixel online calculations of box-AMFs would avoid interpolation errors; Castellanos et al. (2015) estimated the differences between on-line and LUT-derived AMFs to be on average less than 1%, for individual measurements less than 8% , with an upper bound of the difference of 20% over South America.





### 3.3.1 Harmonized settings

Four groups used the same settings (forward model parameters, a priori profiles, temperature and cloud correction) to calculate clear sky and total tropospheric $NO_2$ AMFs for one specific OMI orbit over Australia and Eastern Asia on 02 February 2005 (See Fig. 6). The selected harmonized settings were those from KNMI/WUR (see Table 3). All groups applied the same temperature correction (from Boersma et al. (2004), (see Eq. S1)) and cloud correction via the independent pixel approximation. The aim of this comparison was to obtain an estimate of the structural AMF uncertainty introduced by different vertical discretization and the interpolation schemes assuming that the values of the selected forward model parameters are true.

All groups calculate similar AMF spatial patterns for the selected orbit. Figure 6 (upper panels) shows total tropospheric $NO_2$ AMFs calculated by each group. The distribution of the AMF values along the orbit is determined by the different parameters on which AMFs depend. Lower panels in Fig. 6 show $NO_2$ (a priori) model vertical column, surface albedo and cloud fraction in the orbit. At high latitudes, where surface albedo is high, AMFs are up to 3-5. Surfaces with high albedo (usually covered by snow or ice) reflect more radiation than surfaces with lower surface albedo, and this increases the AMF values. The effect of clouds and the a priori profile is also visible: AMFs are generally low in cloudy regions and over polluted regions in east China ($\sim$30°N), indicative of reduced sensitivity to $NO_2$ in the lowest layers of the atmosphere.

The correlation between AMFs calculated by the different retrieval groups is excellent ($R^2 > 0.99$). Overall, tropospheric AMFs calculated by each of the groups agree within 6.5% in polluted areas and within 2.5% in clean remote areas for most retrieval scenarios, in line with the results from the box-AMF LUT comparison. BIRA AMFs are on average higher than AMFs by the other groups, generally by a few percent, and IUP-UB AMFs are on average lower for polluted and unpolluted situations. Table 2 summarizes the results of the comparison.

Largest differences are found at the edges of the OMI orbit, where viewing zenith angles are large and light paths are long. This can be seen in the lower right panel of Fig. 6, where the relative differences of tropospheric $NO_2$ AMFs between MPI-C and WUR are clearly visible at the edges of the orbit. These differences are consistent with the higher sensitivity to tropospheric trace gases for extreme viewing zenith angles (also shown in Fig. 4(e)) in McArtim compared to DAK. Figure S1 in the supplementary material shows the relative differences between all AMFs calculated by the groups. Relative difference distributions show patterns that reflect the spatial distribution of surface albedo, clouds and $NO_2$ (e.g. over southeastern Australia, East China and Korea). Large differences between the groups are found in cloudy situations. These effects reflect the uncertainties arising from the use of different RTM as well as from the interpolation and the vertical discretization of the LUT when calculating the AMFs.

These results demonstrate that even when similar RTMs, box-AMFs, and identical forward model parameters are used to calculate the AMFs, there is structural uncertainty that is introduced by the specific implementation of different groups. First, the choice of a RTM introduces uncertainty in the box-AMF calculation. Second, there are interpolation errors that are intrinsic to the calculation method using Eq. 3, i.e. interpolation errors in finding the AMF value from the 6-D LUT and the vertical discretization of the a priori profile. Overall, the average differences between the AMFs (always below 6.5% for cloud fractions less than 0.2) are somewhat higher than the differences from the LUT comparison (2%). This means that in successive steps



**Table 2.** Statistical parameters for the comparison of total tropospheric $NO_2$ AMFs for polluted and unpolluted pixels (pixels with model $NO_2$ vertical column higher or lower than $1 \cdot 10^{15}$ molec/cm$^2$ respectively) between the different retrieval groups for one complete orbit from 02 February 2005 (2005m0202t0339-o02940 v003). Only pixels with effective cloud fraction $\leq 0.2$ are considered. Mean, median and sigma are relative differences in % (100(a-b)/a).

| Polluted pixels (#1983) | | | | | | |
|---|---|---|---|---|---|---|
| Diff. between | Mean (rel. diff.) | Median (rel. diff.) | $\sigma$ (rel. diff.) | $R^2$ | Slope | Offset |
| IUP-WUR | -3.8 ± 0.3 | -2.5 | 6.4 | 0.9968 | 0.96 | 0.08 |
| BIRA-WUR | 0.5 ± 0.02 | 0.5 | 0.8 | 0.9996 | 0.98 | 0.02 |
| BIRA-IUP | 3.9 ± 0.7 | 2.9 | 4.8 | 0.9967 | 1.02 | -0.07 |
| MPIC-WUR | -1.5 ± 0.1 | -0.9 | 4.7 | 0.9957 | 0.99 | 0.03 |
| MPIC-IUP | 2.1 ± 0.9 | 0.5 | 4.9 | 0.9955 | 1.03 | -0.06 |
| MPIC-BIRA | -2.0 ± 0.1 | -1.2 | 4.7 | 0.9957 | 1.01 | 0.01 |
| Unpolluted pixels (#23744) | | | | | | |
| IUP-WUR | -0.4 ± -0.3 | -0.3 | 2.4 | 0.9983 | 0.96 | 0.06 |
| BIRA-WUR | 0.6 ± 0.004 | 0.3 | 0.8 | 0.9995 | 0.98 | 0.03 |
| BIRA-IUP | 1.0 ± 0.04 | 0.7 | 1.9 | 0.9989 | 1.01 | -0.04 |
| MPIC-WUR | -0.5 ± 0.02 | -0.4 | 2.1 | 0.9985 | 0.97 | 0.06 |
| MPIC-IUP | -0.1 ± 0.06 | -0.4 | 2.2 | 0.9981 | 1.01 | -0.01 |
| MPIC-BIRA | -1.1 ± 0.02 | -0.9 | 1.7 | 0.9990 | 0.99 | 0.03 |

of the AMF calculation sources of systematic uncertainty are added that propagate throughout the AMF calculation process.
These sources directly affect the agreement between the AMF calculated by different groups and hence the AMF structural uncertainty.

### 3.3.2 Cloud correction: IPA vs. cloud masking

It is important to account for the effect of clouds on the photon path lengths in the troposphere when calculating tropospheric AMFs. Various approaches are commonly used to calculate AMFs in (partly) cloudy situations: (1) the independent pixel ap-
proximation (IPA), introduced in Eq. (4) (e.g. Martin et al. (2002)), and (2) the cloud masking approach (CM) which considers that clear-sky AMFs are a good approximation for scenes with a sufficiently small cloud fraction (e.g. Richter and Burrows (2002)). An important motivation for using the IPA is that few pixels are completely cloud-free. Many pixels still have some degree of cloud cover, and even small cloud fractions strongly affect the sensitivity to the trace gas. The relevant physical effect of clouds (reduced sensitivity to trace gas below the cloud and enhanced sensitivity to trace gas above and in the top layer of
the cloud) is explicitly taken into account in the IPA. The motivation for using cloud masking instead of IPA is that for scenes with small cloud fractions (e.g. < 0.2), retrieved cloud parameters (cloud fraction and cloud pressure) have relatively high





**Figure 6.** Upper panels: total $NO_2$ tropospheric AMFs calculated by BIRA, IUP-UB, WUR and MPI-C. Lower panels: $NO_2$ model tropospheric vertical column (from a priori TM4 profile), climatological surface albedo (from Kleipool et al., 2008), cloud fraction (from $O_2$-$O_2$ and FRESCO+) and an example of the relative differences between MPI-C and WUR AMFs. Only pixels for SZA < 70° are shown. The selected OMI orbit is from 02 February 2005 (2005m0202-o02949-v003).

uncertainty. This inhibits the reliable modeling of the effect of clouds on photon path lengths, and consequently, a clear-sky AMF is used in the CM approach.

To quantify the differences between the two approaches, we compare here tropospheric $NO_2$ AMFs calculated with the IPA approach and with the CM approach for two complete days of OMI measurements (02 February 2005 and 16 August 2005). In polluted situations, IPA AMFs are smaller than CM AMFs, with differences as large as -40% for cloud fractions approaching the threshold value of 0.2 (left panel of Figure 7). The negative differences between IPA and CM are largest for the highest clouds, illustrating the reduced sensitivity to tropospheric $NO_2$ below the cloud in the IPA. IPA AMFs are larger than cloud-




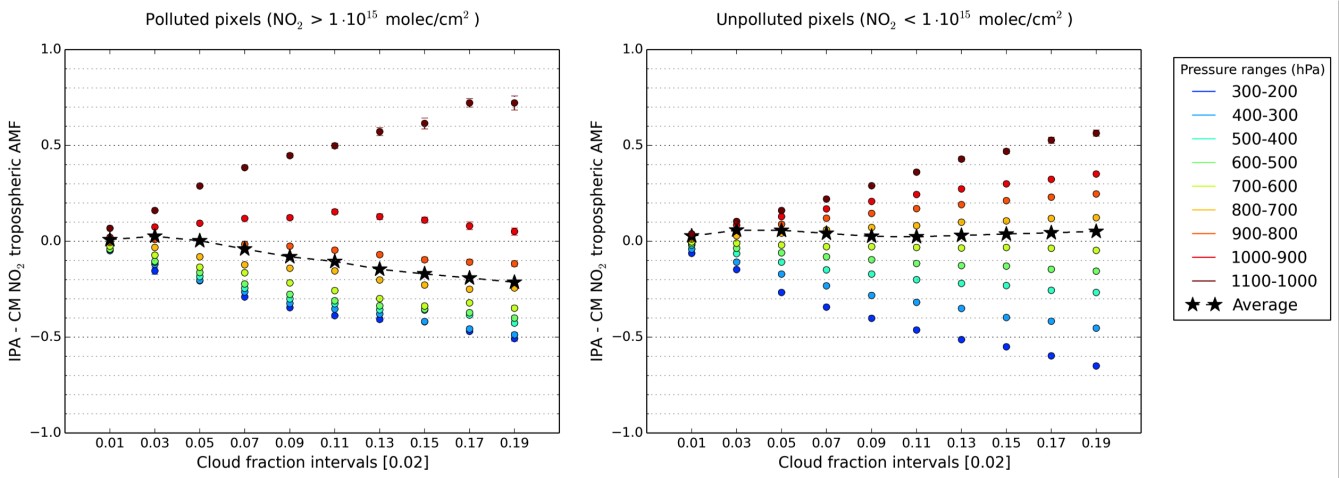

**Figure 7.** Mean absolute differences between IPA and cloud masking (CM) NO$_2$ tropospheric AMFs for different cloud fraction intervals at different cloud pressures ranges (different colors) for a complete day of OMI measurements (02 February 2005). Left panel is for polluted situations and right panel for unpolluted situations (pixels with model NO$_2$ vertical column higher or lower than $1 \cdot 10^{15}$ molec/cm$^2$ respectively). The stars with the black dashed lines show the average difference for all the cloud pressures. Pixels with surface albedo less than 0.3 and SZA < 70° are considered.

masking AMFs for clouds situated in the lower troposphere (cloud pressure > 900 hPa), where most NO$_2$ pollution resides.

These positive differences can be understood from the albedo effect of residual clouds. Low, bright clouds lead to enhanced photon scattering through the NO$_2$ layers above the cloud level and also inside the cloud top layer, and this increases the sensitivity to NO$_2$. For polluted situations, IPA AMFs are on smaller than CM AMFs by 20% for cloud fractions of 0.05-0.2, and smaller by 11% for cloud fractions between 0.0-0.2.

  In unpolluted situations, IPA and CM AMFs are generally quite similar, with average relative differences within 5%. Still,

there are important differences between the two approaches. In unpolluted situations with clouds in the free and upper troposphere (cloud pressure < 600 hPa), IPA AMFs are smaller because of reduced sensitivity to NO$_2$ (right panel of Figure 7). For clouds in the lower troposphere, IPA AMFs are larger because of the albedo effect. The change of sign in the differences between IPA and CM AMFs now occurs near 700 hPa (instead of near 900 hPa for polluted scenes), reflecting the more even vertical distribution of NO$_2$ in pristine situations compared to polluted scenes when most NO$_2$ resides in the polluted boundary

layer.

  These results indicate that the differences between IPA and CM AMFs are substantial especially for polluted situations and small residual cloud fractions. Selecting a particular cloud correction approach implies that AMF values that will be systematically different from values obtained with the other method. In polluted situations, the differences are 20-40% for cloud fractions between 0.1-0.2, with cloud pressure largely explaining the magnitude and sign of the differences. Note that

the a priori profiles used to calculate the AMFs in this section have been obtained from a specific CTM. If a different CTM





was used, the values for the differences between IPA and CM AMFs would be different, in line with the structural uncertainty that is being discussed in this study (See Sect. 3.3.3). A previous study by Van Noije et al. (2006) reported 30% higher GOME tropospheric $NO_2$ columns retrieved using the IPA compared to retrievals using the cloud-masking approach. Such differences are in line with the systematically lower IPA AMFs found here. But, like the study by Van Noije et al. (2006), we cannot

clearly recommend one AMF approach over the other. In order to make such a recommendation, a more detailed analysis of the cloud parameter uncertainties is needed, along with a validation of tropospheric $NO_2$ retrievals using different AMF approaches against independent reference data. Such a validation exercise should preferably focus on polluted situations with small (0.05-0.2) residual cloud fractions.

### 3.3.3    Round robin comparison

For the round robin comparison, each group calculated tropospheric $NO_2$ AMFs using their preferred settings (i.e. their own preference for source of forward model parameters, cloud and aerosol correction). We extended the comparison and included other leading international retrieval groups (University of Leicester, NASA and Peking University). We now have a wider range of approaches and assumptions to better evaluate the impact that the calculation methods and choices of forward model parameters have on the structural uncertainty.

Table 3 summarizes the AMF algorithms included in this comparison. There are several differences with the harmonized settings used in the previous section. IUP-UB and BIRA now apply IPA only when cloud fraction is less than 0.1 and 0.2, respectively, motivated by the high uncertainty of cloud parameters for scenes with small cloud fractions (see Sect. 3.3.2). Peking University accounts for the surface reflectance anisotropy and they do pixel-by-pixel online radiative transfer calculations. They also include an explicit aerosol correction, motivated by the fact that the implicit aerosol correction breaks down

in situations of high aerosol optical thickness and strongly absorbing particles (Castellanos et al. (2015), Chimot et al. (2016)), which is particularly significant in East China. MPIC applies IPA cloud correction for clouds higher than 3km and cloud masking for clouds between 2 and 3 km when cloud fraction is less than 0.1. For clouds below 2 km they include a parameterized aerosol-cloud layer in order to account for the possibility of cloud aerosol mixtures, which might be especially relevant for AMF calculation in scenarios where trace gas is most abundant in the lowest part of the troposphere. Among all the groups,

five different chemistry transport models for the a priori $NO_2$ profiles are used.

The agreement of AMFs from this round robin exercise quantifies the overall AMF structural uncertainty. The comparison with 7 groups allowed us to calculate a mean AMF as a reference (which is not necessarily the true AMF) value which can be considered a state-of-the-art AMF value. For a representative ensemble mean AMF, we required all groups to have a valid (unflagged) AMF value at a pixel location. We selected two different days (02 February 2005 and 16 August 2005) in winter

and summer to identify possible seasonality effects in the agreement of the AMFs.

**Round robin: identical cloud parameters**

First we compare the 6 groups that use the same cloud parameters. In contrast to what we found in the harmonized settings comparison, the global maps of tropospheric AMF calculated by each group using their preferred settings (Fig. 8) show pro-





**Table 3.** Overview of AMF calculation methods and ancillary data used in the round robin experiment by various research groups.

| Group and reference | RTM | LUT interpolation | Surface reflectivity | Surface pressure | Cloud parameters | Cloud correction | Aerosol correction | A priori profile |
|---|---|---|---|---|---|---|---|---|
| BIRA-IASB (Sect. S1.1) | VLIDORT | Linear in 6D space | MODIS BSA and OMI Min LER | GMTED2010* | $O_2$-$O_2$ | IPA: CF>0.2 CM: CF<0.2 | Implicit | Daily TM5 (1° x 1°) |
| IUP-UB (Sect. S1.2) | SCIATRAN | Linear in 6D space | Min LER Kleipool et al. (2008) (v003) | GMTED2010* gridded to 0.25° x 0.25° | $O_2$-$O_2$ | IPA: CF>0.1 CM: CF<0.1 | Implicit | MACC-II daily reanalysis (1.125° x 1.125°) |
| KNMI/WUR Boersma et al. (2011) | DAK v3.31 | Linear in 6D space | Min LER Kleipool et al. (2008) (v002) | Global 3km DEM** Pixel average | $O_2$-$O_2$ | IPA | Implicit | Daily TM4 (3° x 2°) |
| Uni. Leicester Barkley et al. (2011, 2012, 2013) | LIDORTv2.3 | Linear in 4D space | Mode LER Kleipool et al. (2008) (v002) | GEOS-Chem surface pressure | $O_2$-$O_2$ | IPA | Implicit | Daily GEOS-Chem (2° x2.5°) |
| MPI-C (Sect. S1.3) | McArtim | Linear in 6D space | Min LER Kleipool et al. (2008) (v002) | Global 3km DEM*** Pixel average | $O_2$-$O_2$ | IPA > 3km Cloud masking between 2-3 km | Explicit*** for clouds below 2km | Daily TM4 (3° x 2°) |
| NASA-GFSC Bucsela et al. (2013) Lamsal et al. (2014) | TOMRAD | Linear in 6D space | Min LER Kleipool et al. (2008) (v002) | Global 3km DEM Pixel center | $O_2$-$O_2$ | IPA | Implicit | Monthly mean GMI (2.5° x 2°) |
| Peking Uni. Lin et al. (2014, 2015) | LIDORT v3.6 | Online calculations | MCD43C2 BRDF | GEOS-Chem (0.5° x 0.667°) | POMINO retrieval | IPA | Explicit GEOS-Chem daily AOD | Daily GEOS-Chem (0.5° x 0.667°) |

*Global Multi-resolution Terrain Elevation Data.

**Digital Elevation Model data.

***See Sect. S1 in the supplement for more detailed information.

nounced differences in several regions. For example, over the Sahara desert, where surface albedo is high (see lower panel on
Fig. 8), AMFs differ by up to 15%. Small differences in the albedo values can lead to high differences in the AMFs, especially
for surface albedos lower than 0.3 (see Fig. 4 (b)). Over Central Africa, AMFs differ in situations where cloud fraction is close
to the typically applied threshold of 0.2 (left lower panel in Fig. 8).

We compared global AMF calculations from all individual groups against the pixel mean AMF from 6 groups (Peking
University only calculates AMFs over China). Figure 9 shows the average ratio of the AMF by each group to the ensemble mean
AMF (bars) and the correlation (crosses) for polluted situations ($NO_2 > 1^{15}$molec/cm$^2$, left panel) and unpolluted situations
($NO_2 < 1^{15}$molec/cm$^2$, right panel). Over polluted regions, the agreement among the 6 groups is within 12-42% in February and
within 10-31% in August. BIRA AMFs are 14% higher than the ensemble mean, and WUR AMFs are 18% lower, suggesting
considerable structural uncertainty. Over unpolluted regions the agreement is better: AMFs from the different groups agree
within 8.5-18% in both February and August, which implies a smaller structural uncertainty (Table S6 provides a detailed
summary of the comparison).

In order to asses which forward model parameters explain most of the AMF structural uncertainty, we analyzed AMF differ-
ences from groups that use identical cloud parameters and implicit aerosol correction (BIRA, University of Leicester, NASA
and WUR). Between these four groups, the only different forward model parameters are surface albedo, a priori $NO_2$ profile



and surface pressure. To investigate which of these parameters best explains the AMF variability, we correlated differences
between a particular parameter ($\Delta A_s$, $\Delta NO_2$ and $\Delta P_s$) with the corresponding AMF differences ($\Delta AMF$). For each particu-
lar parameter, we required the differences in the other parameters to be small (surface albedo within $\pm 0.02$, surface pressure
within $\pm 50$ hPa and a priori $NO_2$ vertical columns within $\pm 0.2 \, 10^{15}$ molec/cm$^2$) so we could isolate the effect of one parameter
only, while keeping sufficient pixels for statistical significance.

We focus on explaining the differences between BIRA and WUR here, since these were on the order of 30% (Fig. 9). We
explored the correlations between BIRA-WUR AMF differences and differences between assumed surface pressures, albedos,
and $NO_2$ vertical columns and profile shapes; results are shown in Fig. S3 and Table S3. We find that surface pressure differences
do not explain the large systematic AMF differences, and that surface albedo differences explain some of the WUR and BIRA
AMF differences, mostly in winter, when $NO_2$ is found close to the surface and AMFs are more sensitive to albedo variations
than in summer. The WUR-BIRA AMF differences however are highly sensitive to the differences between the a priori $NO_2$
profiles used. $NO_2$ profiles are vertically more elevated in TM5 (used by BIRA) than in TM4 (used by WUR) (right panel of
Fig. S3), as diagnosed by their 20 hPa lower effective $NO_2$ pressures (pressure levels weighted by $NO_2$ sub-column in that
level). The confinement of the trace gas to lower atmospheric layers and the higher concentrations explains the systematically
lower AMF values for WUR compared to BIRA.

Selecting a specific chemistry transport model thus influences the AMF structural uncertainty via differences in the profile
shape. These differences in the profile shape depend on the different characteristics of the models (e.g. spatial and temporal
resolution and parameterization of different processes in the atmosphere). PRevious studies analysed how using different CTMs
influences the $NO_2$ retrievals due to the change in the profile shapes used to calculate the AMF values. Heckel et al. (2011)
compared retrievals using fine and coarse resolution models and concluded that using one AMF value for a large heterogeneous
scene can lead to 50% bias in the retrieved NO2 columns.Vinken et al. (2014) reported much smaller average differences of
10% in retrieved $NO_2$ columns mainly due to different emission inventories used in TM4 (3° x 2°) and WRF-Chem (0.5° x
0.67°). According to (Laughner et al., 2016), different temporal resolution also influences a priori profile shapes; they found
differences in the retrieved $NO_2$ column for individual days up to 40% that were mostly explained by day-to-day wind direction
variations that were not captured in the monthly averages.

All these aspects influence the estimation of retrieval (and AMF) theoretical uncertainties. In order to quantitatively estimate
the effect of one model characteristic alone (e.g. the spatial resolution) on the AMF structural uncertainty it would be necessary
to compare AMF calculated with the same approach but with just that specific characteristic being different in the profile shapes
generated by the CTM. Such specific sensitivity analysis has not been done in this study but should be considered in future
AMF comparisons. The findings in this subsection indicate that quality assurance efforts for $NO_2$ retrievals should not focus
just on column validation, but also target the validation of the a priori $NO_2$ profiles used in the AMF calculations. It is worth
to note that using the averaging kernels will reduce the effect of the a priori trace gas profiles chosen in the retrieval scheme.



**Round robin: different cloud parameters**

In the previous section, we found that differences between a priori $NO_2$ profiles are the main cause for AMF structural uncertainty when cloud parameters are identical in AMF calculation approaches. Here we extend our round robin experiment by including AMF calculations from Peking University (Lin et al. (2014, 2015)) that were done with different cloud parameters

(Table 3) than the $O_2$-$O_2$ cloud parameters used by all other groups. The comparison of Peking University and WUR AMFs thus allowed us to investigate the relative importance of differences in cloud parameters in driving AMF structural uncertainty. Our comparison of AMFs is confined to China, since Peking University calculations are only available over that region.

All the groups calculate similar spatial patterns for the AMFs over China (Fig. 10). In the polluted northeast (Beijing area) the AMFs are lower due to the reduced sensitivity to $NO_2$ in the lower troposphere. In the western part over the Tibet region,

AMFs are higher due to the presence of ice and snow in February. Figure 11 shows the average ratio of each group's AMF to the ensemble mean AMF (bars) and the correlation (crosses) for polluted situations (left panel) and unpolluted situations (right panel). In polluted regions, AMFs generally agree within 37% in February and within 20% in August, and correlations are 0.7-0.9. Peking University AMFs are higher than the ensemble mean AMF, especially in August when they are 25% higher. WUR and MPI-C AMFs are lower than the mean AMF, especially in August (20% lower). In unpolluted regions the agreement

is better: within 26% in February and within 16% in August, with correlation of 0.8-0.95 (see Table S7).

To estimate the effect of differences in cloud parameters on AMF structural uncertainty, we analyzed differences in AMF calculated by WUR and Peking University. The Peking University AMF calculations (and the cloud parameters) were based on a version of the POMINO retrieval using clouds retrieved with an implicit aerosol treatment (i.e. similar to KNMI/WUR). We explored the correlations between Peking University and WUR AMFs differences and differences in cloud pressure ($P_c$)

and $NO_2$ vertical columns by requiring the differences in other forward model parameters to be relatively small. Results are shown in Fig. S4 and Table S4. AMF differences are partly explained by differences in the effective cloud pressures (Table S4): the $O_2$-$O_2$ cloud pressures used by WUR are systematically lower (by 100 hPa) than those by Peking University, in line with Veefkind et al. (2016). This results in stronger screening of below-cloud $NO_2$ pollution, and consequently lower AMFs by WUR compared to Peking University AMFs. Peking University uses $NO_2$ profiles from GEOS-Chem. These profiles tend

to peak at higher vertical levels than those from TM4 (Lin et al. (2014), Boersma et al. (2016)), thus contributing to higher AMFs by Peking University compared to WUR AMFs. In summary, the more elevated $NO_2$ profiles in combination with less elevated clouds explain the substantially higher AMF by Peking University than WUR AMFs.

**Round robin: explicit aerosol correction**

The POMINO retrieval by Peking University explicitly corrects for the presence of aerosols in the atmosphere by including

profiles of aerosol optical properties simulated by the GEOS-Chem model (and constrained by MODIS AOD on a monthly basis) in the radiative transfer model and in the cloud retrieval (Lin et al. (2014, 2015)). Most of the other groups (see Table 3) assume that the aerosol effects are implicitly accounted for in the cloud retrievals (Boersma et al. (2011), Castellanos et al. (2015)). Including an explicit aerosol correction influences AMF values indirectly by changes in cloud fraction and cloud





pressure and directly in the radiative transfer simulations. We quantify the effect of the choice of aerosol correction in AMF
structural uncertainty by comparing AMFs calculated by Peking University with (abbreviated $AMF_{aer}$ hereafter) and without
(AMF) explicit aerosol correction.

   In situations with substantial aerosol pollution (AOD > 0.5), selection of one aerosol correction approach over the other can
result in AMF structural uncertainty of 45% over China. The sign of the AMF differences depends mainly on the altitude of
the aerosol layer relative to the $NO_2$ profile (see e.g. Leitao et al. (2010)). We find that $AMF_{aer}$ are on average 55% smaller
in situations when aerosols are located above the $NO_2$ layer, mainly because cloud pressures are lower on average (more than
350 hPa), resulting in stronger screening of $NO_2$ (upper panel of Fig. S5; Table S5). When the aerosol vertical distribution
is similar to that of $NO_2$, $AMF_{aer}$ are on average 45% higher, mostly because of much smaller cloud fractions, resulting in
reduced screening of below-cloud $NO_2$ (lower panel in Fig. S5; Table S5). An additional factor is that when aerosols are mixed
with $NO_2$, they increase the optical light path and enhance AMF values. These results are in line with Lin et al. (2015) where
an evaluation of the influence of the aerosols in the $NO_2$ retrieval is analyzed for 2012.

### 3.4  Stratospheric air mass factors

We pointed out in Sect. 3.2 that differences in the description of the atmosphere's sphericity could lead to differences in
stratospheric AMFs, especially for extreme geometries. Here we investigate the differences between stratospheric $NO_2$ AMFs
calculated with DAK and McArtim radiative transfer models. The McArtim model simulates the radiative transfer in an at-
mosphere that is spherical for incoming, single-scattered, and multiple-scattered light. DAK's atmosphere is spherical for
incoming sunlight, but plane-parallel for scattered sunlight. Based on these differences, we may expect the average photon
paths at high altitudes in McArtim to be shorter than in DAK, as diffuse photon contributions (from near-horizontal directions)
in McArtim are bound to finite spherical atmosphere (as illustrated in Figure S2). Consequently, stratospheric AMFs in McAr-
tim are smaller (Fig. 4(a)). Figure 12 shows that McArtim box-AMFs (at 25 hPa) are systematically lower than those from
DAK by 1-2% for moderate viewing geometries, with more significant differences (up to -5% to -10%) when solar zenith and
viewing angles are large.

   A direct validation of stratospheric $NO_2$ AMFs is difficult, but comparing simulated stratospheric slant column densities
against observed $NO_2$ SCDs constitutes a test of the radiative transfer models. Here we use OMI-observed (un-destriped)
SCDs over the Pacific from the OMNO2A v1 product (van Geffen et al. (2015), Boersma et al. (2011)) as benchmark. The
$NO_2$ columns over the Pacific Ocean are dominated by stratospheric $NO_2$, so we expect simulated stratospheric SCD values to
be similar or somewhat smaller than the observed, total SCDs. Simulated SCDs are the product of modelled VCDs (from data
assimilation in TM4) and the stratospheric AMFs calculated with DAK and McArtim. Figure 13 (left panel) indicates (for high
solar and viewing zenith angles) that stratospheric SCDs simulated with McArtim are close to, or slightly below the OMI SCDs.
In contrast, the stratospheric SCDs simulated with DAK overtop the OMI SCDs, because of the higher stratospheric AMFs
from that model. This inevitably leads to negative values for SCD-$SCD_{strat}$, and consequently to reduced or even negative
tropospheric $NO_2$ VCDs at high latitudes. Indeed, DOMINO v2 retrievals (using DAK stratospheric AMFs) are known to
suffer from negative tropospheric VCDs at high latitudes especially in the summer hemisphere (Beirle et al., 2016) when solar



zenith angles are largest. For small solar zenith angles in the Tropics, the differences between DAK and McArtim stratospheric slant columns are smaller, but still appreciable at the edges of the swath (Fig. 13 (right panel)).

We tested whether possible errors in the diurnal cycle of stratospheric $NO_2$ could explain the overestimated slant columns for extreme viewing geometries. We did so by imposing stratospheric $NO_2$ vertical columns that are either constant with OMI row number (i.e. with local time), or increase (as $N_2O_5$ photolysis, $NO_2$ concentrations build up) at a rate of approximately $0.15\ 10^{15}$ molec/cm$^2$ h$^{-1}$, i.e. by $1\ 10^{15}$ molec/cm$^2$ from the left to the right side of the orbit (Fig. S6(a)). These estimates correspond to the range of increase rates at high latitudes in summer reported in the literature (e.g. Vaughan et al. (2006),

Celarier et al. (2008), Dirksen et al. (2011)). Our tests show that for these scenarios, simulated SCDs based on McArtim generally stay within the observational constraints of the OMI SCD patterns but that the simulated SCDs based on DAK are still exceeding the observed SCDs (Fig. S6(b)-(c)). McArtim provides a better physical description of photon transport in the stratosphere. The results above are not yet fully conclusive; a complete test would require the implementation of McArtim (instead of DAK) in the data assimilation scheme, or a dedicated validation of $NO_2$ columns with independent reference data

in situations with extreme viewing geometries. Nevertheless, our results clearly hint at McArtim as the RTM providing the more realistic stratospheric AMFs, and we will test this assumption further in the remainder of the QA4ECV project.

## 4    Conclusions and recommendations

We have analysed in detail the AMF calculation process for $NO_2$ and HCHO satellite retrievals from seven different retrieval groups. By comparing approaches for every step of the AMF calculation process we have identified the main sources of

structural uncertainty and we have traced back these uncertainties to their underlying causes. We have estimated the structural uncertainty in the $NO_2$ AMF calculation, which results from methodological choices and from preferences and assumptions made in the calculation process. Structural uncertainty is relevant beyond theoretical algorithm uncertainty, which typically only addresses the propagation of errors within the context of one particular retrieval algorithm.

The choice of RTM for TOA reflectance and box-AMF calculation introduces an average uncertainty of 2-3%. The de-

tailed comparison showed that state-of-the-art RTMs are in good agreement. Particularly for DAK, this is the first time that box-AMF calculations are extensively tested against those calculated with other RTMs. The McArtim model simulates systematically lower box-AMFs in the stratosphere, which we attribute to the model's geometrically more realistic description of photon scattering in a spherical atmosphere. The four European retrieval groups agree within 6% in their calculation of $NO_2$ tropospheric AMFs when identical ancillary data (surface albedo, terrain height, cloud parameters and a priori trace gas profile)

and cloud correction are used. This demonstrates that the selection of RTM and the interpolation operations lead to modest uncertainty, which is intrinsic to the calculation method chosen and therefore cannot be avoided.

When retrieval groups use their preference for ancillary data along with their preferred cloud and aerosol correction, we find that the structural uncertainty of the AMF calculation is 42% over polluted regions and 31% over unpolluted regions. Table 4 shows the escalation of the structural uncertainty with every step of the AMF calculation. The steep increase from 6% to

42% strongly suggests that it is not the models or the calculation method but the assumptions and choices made to represent



the state of the atmosphere that introduces most structural uncertainty in the AMF calculation. The structural uncertainty is of similar magnitude as the theoretical uncertainties found in algorithm error propagation studies which confirms that there is a substantial systematic component in trace gas satellite retrieval uncertainties.

**Table 4.** Average relative structural uncertainty for every step of the AMF calculation following the comparison process shown in Fig. 1. This includes the modelling of TOA reflectance ($\sigma_R$), calculation of box-AMF LUT ($\sigma_m$), tropospheric AMFs using harmonized settings ($\sigma_M$) and the overall structural uncertainty from AMF using preferred settings ($\sigma_{M'}$).

|  | $\sigma_R$ | $\sigma_m$ | $\sigma_M$ | $\sigma_{M'}$ |
|---|---|---|---|---|
| **NO$_2$** | 1.1% | 2.6% | 6% | 31% - 42% |
| **HCHO** | 1.5% | 2.6% | | |

Sensitivity studies for one particular algorithm indicate that the choice for cloud correction (IPA or cloud masking) is a strong source of structural uncertainty especially for polluted situations with residual cloud fractions of 0.05-0.2 (on average an structural uncertainty of 20%). The choice for aerosol correction (explicitly or implicitly via the cloud correction) introduces an average uncertainty of 50%, especially when aerosol loading is substantial. Selecting trace gas a priori profiles from different chemistry transport models, surface albedo from different datasets and cloud parameters from different cloud retrievals contributes substantially to structural uncertainty in the AMFs. These findings point to the need for detailed validation experiments designed to specifically test cloud and aerosol correction methods under relevant conditions (strong pollution, residual cloud fractions of 0.1-0.2). Not just the retrieved NO$_2$ column itself should be validated, but also the a priori vertical NO$_2$ profile, the cloud and aerosol distributions, and the surface albedo values should be compared in detail against independent reference measurements.

The magnitude of the structural uncertainty in AMF calculations is significant, and is caused mainly by methodological differences and particular preferences for ancillary data between different retrieval groups. This study provides evidence for the need of improvement of the different ancillary datasets, including uncertainties of the forward model parameters used in the retrievals for a better agreement in the AMF calculation. This will decrease significantly AMF structural uncertainty towards the levels desired in user requirement studies ($\pm$ 10 %). As there is no "true" AMF value to be used as reference, it is difficult to decide which approach and which ancillary data are the best. For this reason, future research should include a thorough validation against independent reference data, specifically in the situations where AMF structural uncertainty has highest impact.

*Acknowledgements.* This research has been supported by the FP7 Project Quality Assurance for Essential Climate Variables (QA4ECV), no. 607405. A.H. and A.R. acknowledge funding by DLR in the scope of the Sentinel-5 Precursor verification project (grant 50EE1247). UoL acknowledges the use of the ALICE and SPECTRE High Performance Computing Facility at the University of Leicester.



**Figure 8.** Tropospheric NO₂ AMFs calculated by each of the groups for a complete day of OMI measurements (02 February 2005). Lower panels show an example of cloud fraction and surface albedo used by KNMI/WUR (showed as example; see Table 3) to calculate the AMFs. Groups apply different filters to the measurements which explains the different gaps (grey).




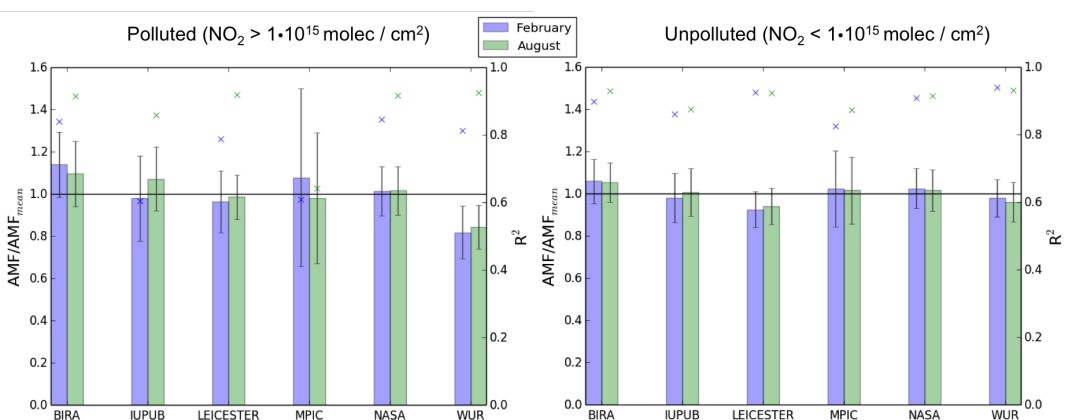

**Figure 9.** Ratio of tropospheric $NO_2$ AMFs by each group to the ensemble mean (left axis, bars) and the correlation coefficient (right axis, cross) for two complete days of OMI measurements (02 February 2005 (blue) and 16 August 2005 (green)) over the globe for polluted (left panel) and unpolluted (right panel) pixels. The error bars correspond to the standard deviation. Only pixels for SZA < 60 ° and cloud fraction < 0.2 are considered in the analysis.





**Figure 10.** Tropospheric NO$_2$ AMFs calculated by each of the groups for a complete day of OMI measurements (02 February 2005) over China ( 20°N-53°N/ 80°W-130°W). Only pixels for SZA < 60 °, effective cloud fraction < 0.5 and surface albedo < 0.3 are shown.



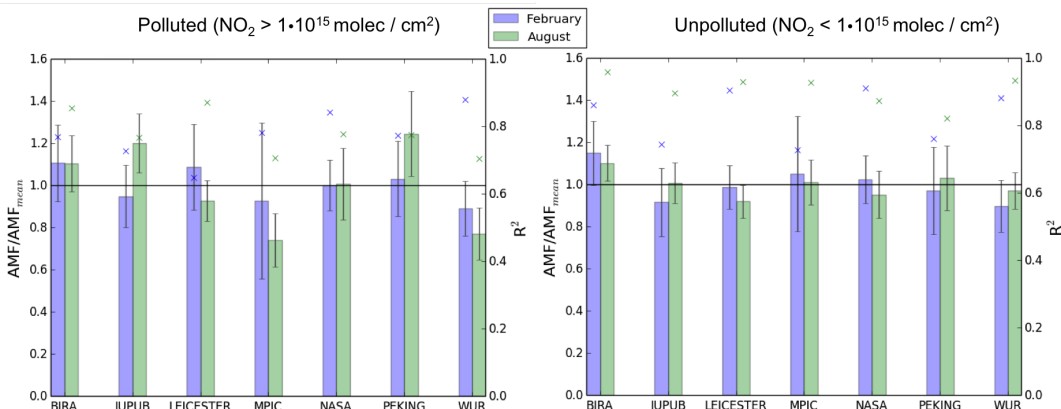

**Figure 11.** Ratio of tropospheric $NO_2$ AMFs by each group to the ensemble mean (left axis, bars) and the correlation coefficient (right axis, cross) for two complete days of OMI measurements (02 February 2005 (blue) and 16 August 2005 (green)) for polluted (left panel) and unpolluted (right panel) pixels over China. The error bars correspond to the standard deviation. Only pixels for SZA < 60 ° and cloud fraction < 0.2 are considered in the analysis.

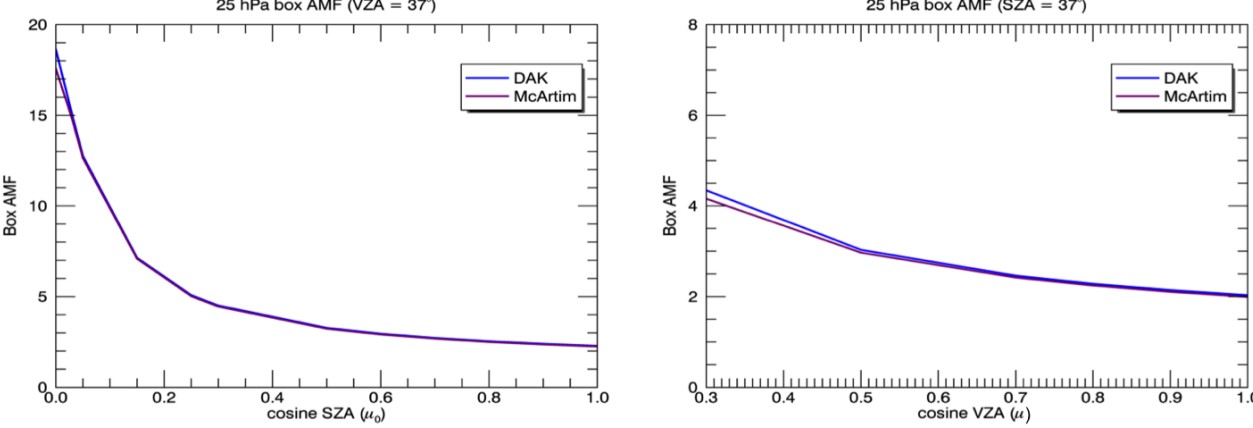

**Figure 12.** Box-AMFs at 25 hPa as a function of cosine of SZA (left panel) and as a function of cosine of VZA (right panel). In the left panel, VZA is constant at $37°$ ($\mu = 0.8$), and at the right panel, SZA is constant at $37°$ ($\mu_0 = 0.8$).





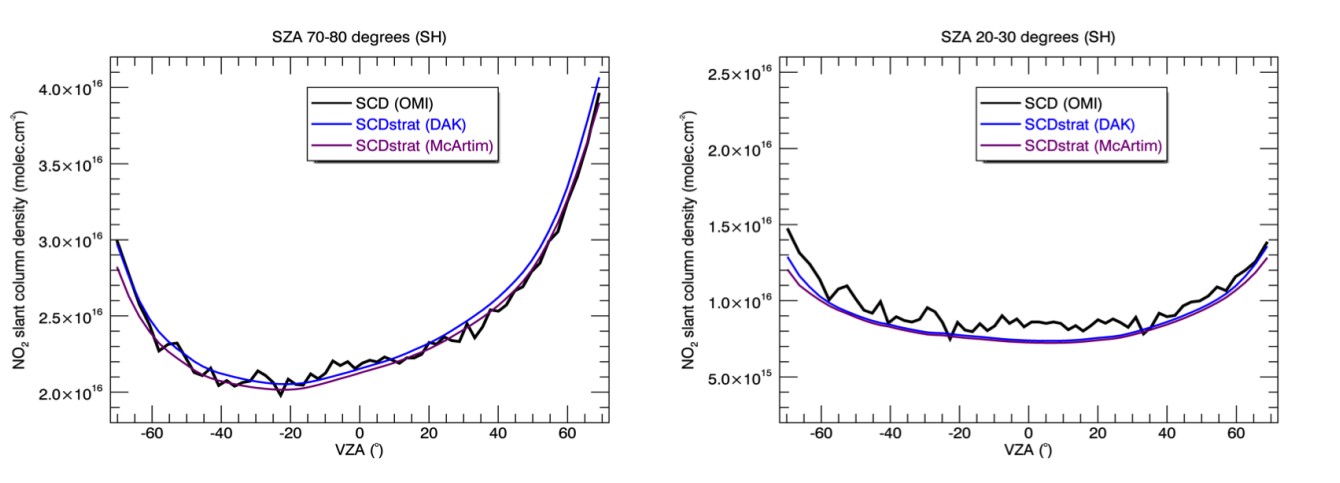

**Figure 13.** Averaged OMI total NO$_2$ SCD (black line) as a function of viewing zenith angle for solar zenith angles between 70-80 degrees (left panel) and 20-30 degrees (right panel) (OMI orbit 02940 on 02 February 2005). The blue line indicates the estimated stratospheric SCDs based on DOMINO v2 stratospheric VCDs and DAK stratospheric AMFs, and the purple line represents the stratospheric SCDs based on DOMINO v2 stratospheric VCDs and McArtim stratospheric AMFs. The only difference between the DAK and McArtim-based stratospheric slant columns is the use of the radiative transfer model; all other relevant parameters (TM4 assimilated stratospheric column, cloud parameters, albedo, NO$_2$ profile shape) are identical.





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
