# Peer review of "Structural uncertainty in air mass factor calculation for NO2 and HCHO satellite retrievals"

_Atmospheric Measurement Techniques, 2016_

## Referee Comment (RC1) · Anonymous Referee #2 · 11 Nov 2016

This paper is interesting and should be published after attention to the items below.

Since the structural error is just the 1-sigma variation among the retrievals, I wonder if the fact that so many of them use the Kleipool albedo and fairly coarse models is underestimating the uncertainty due to albedo and profile shape. Thorne et al. (2005) specified that structural uncertainty should be "aggregated over many independent, plausibly constructed datasets. . .." The paper should address whether the uncertainty due to parameters shared among a large percentage of the retrievals used in the study may bias the interpretation to an underestimate of the uncertainty.

Regarding the discussion in the final paragraph of p. 21, the point that validation of the a priori profiles is important is well taken, and I agree that estimating the effect of only the spatial (or temporal) resolution of the chemical transport model on the re-

trieval would require a very specific study. However our understanding of structural uncertainty is that comparing the AMFs calculated by a variety of retrievals allows a characterization of the total error independent of the parametric uncertainty calculations. Given that the highest resolution a priori profiles used here were the 0.5 x 0.667 degree profiles in the POMINO retrieval, and that Valin et al. (2011), Heckel et al. (2011), and Yamaji et al (2014) all show that model resolution < 20 km is necessary to capture the nonlinearity of NOx chemistry; that Russell et al. (2011), McLinden et al. (2014), and Kuhlmann et al. (2015) show profiles at <= 15 km resolution significantly change the AMF, and Vinken et al. (2014) used a sub-grid plume parameterization in their retrieval with 0.5 x 0.667 degree profile resolution to a similar effect, my concern is that the overall uncertainty in the AMF derived from a structural uncertainty that does not include any retrievals using profiles with < 20 km resolution misrepresents the true uncertainty and bias. If adding at least one retrieval with < 20 km resolution a priori profile is impractical at this point, then at a minimum an extended discussion of the likelihood that the AMF uncertainty derived here is underestimated should be developed.

The final statement on p. 21: "It is worth to note that using averaging kernels will reduce the effect of the a priori trace gas profile chosen in the retrieval scheme." requires additional discussion. My understanding of the use of averaging kernels is that they are useful in two ways:

1) When comparing satellite retrieved VCDs against a model, applying the AKs to the model effectively "retrieves" the model trace gas profile, thus the dependence on the a priori profile in the retrieval is the same for both the observed and modeled column, and cancels out (Eskes and Boersma, 2003).

2) Alternately, one could use AKs to implement one's own a priori profiles in the retrieval.

Only in the first case would I say that the dependence on the trace gas profile is reduced, and that only applies when comparing to a model. Work using the satellite columns directly (e.g. Duncan et al. 2010, Beirle et al. 2011, Valin et al. 2013, Mebust and Cohen 2014, Lu et al. 2015, Liu et al. 2016, etc.) would not be able to use AKs in this way.

Technical corrections:

p. 21 l. 476 - PRevious (the R should be lowercase) p. 21 l. 479 - NO2 (the 2 should be subscript) p. 21 l. 480 - (Laughner et al. 2016) —> Laughner et al. (2016)

References:

Beirle et al., Megacity Emissions and Lifetimes of Nitrogen Oxides Probed from Space, Science, 333, 1737-1739, 2011.

Duncan et al., Application of OMI observations to a space-based indicator of NOx and VOC controls on surface ozone formation, Atmos. Environ., 44, 2213-2223, doi: 10.1016/j.atmosenv.2010.03.010, 2010.

Eskes, H. J. and Boersma, K. F.: Averaging kernels for DOAS total-column satellite retrievals, Atmos. Chem. Phys., 3, 1285-1291, doi:10.5194/acp-3-1285-2003, 2003.

Heckel et al., Influence of low spatial resolution a priori data on tropospheric NO2 satellite retrievals, Atmos. Meas. Tech., 4, 1805-1820, doi: 10.5194/amt-4-1805-2011, 2011.

Kuhlmann et al., Development of a custom OMI NO2 data product for evaluating biases in a regional chemistry transport model, Atmos. Chem. Phys., 15, 5627-5644, doi:10.5194/acp-15-5627-2015, 2015.

Liu et al., NOx lifetimes and emissions of cities and power plants in polluted background estimated by satellite observations, Atmos. Chem. Phys., 16, 5283–5298, doi: 10.5194/acp-16-5283-2016, 2016. Lu et al., Emissions of nitrogen oxides from US urban areas: estimation from Ozone Monitoring Instrument retrievals for 2005–2014, Atmos. Chem. Phys., 15, 10367-10383, doi: 10.5194/acp-15-10367-2015, 2015. McLinden et al., Improved satellite retrievals of NO2 and SO2 over the Canadian oil sands and comparisons with surface measurements, Atmos. Chem. Phys., 14, 3637-3656, doi:10.5194/acp-14-3637-2014, 2014.

Mebust and Cohen, Space-based observations of fire NOx emissions coefficients: a global biome-scale comparison, Atmos. Chem. Phys., 14, 2509-2524, doi: 10.5194/acp-14-2509-2014
, 2014.

Russell et al., A high spatial resolution retrieval of NO2 columns densities from OMI: method and evaluation, Atmos. Chem. Phys., 11, 8543-8554, doi:10.5194/acp-11-8543-2011, 2011.

Valin et al., Effects of model resolution on the interpretation of satellite NO2 observations, Atmos. Chem. Phys., 11, 11647-11655, doi:10.5194/acp-11-11647-2011, 2011.

Valin et al., Variations of OH radical in an urban plume inferred from NO2 column measurements, Geophys. Res. Lett., 40, 1856–1860, doi:10.1002/grl.50267, 2013. Vinken et al., Constraints on ship NOx emissions in Europe using GEOS-Chem and OMI satellite NO2 observations, Atmos. Chem. Phys., 14, 1353–1369, 2014. Yamaji et al., Influence of model-grid resolution on NO2 vertical column densities over East Asia, J. Air. Waste. Manage., 64, 436-444, doi:10.1080/10962247.2013.827603, 2014.

---

## Referee Comment (RC2) · Anonymous Referee #1 · 14 Nov 2016

This paper does a good job breaking down sources of uncertainty in AMF calculations, and will be useful to the satellite retrieval community. I have a few minor comments that need addressing before publication:

Line 132: "It is desirable to use as much information as possible retrieved from the satellite instrument itself." I didn't see much in the model descriptions with regards to this statement. What information from the satellite is used, is it the same for all the models, etc.

Line 275: "The agreement in this study is better than previous RTM comparisons..." Any idea why?

Line 357: The differences between online and LUT AMFs would depend on the resolution of the LUT. Since the Castellanos et al (2015) study indicated an 8% error (which is

larger than the error due to RTM treatments you found), it would be worthwhile to have a brief discussion on the LUTs used in the different models and possible differences that may arise.

Section 3.3.2: Which model is used to evaluate the cloud corrections?

Line 467-470: Characterization of the sensitivity of AMF to albedo and to a priori profiles is inconsistent. Surface albedo is said to "explain some" of the difference, while the AMFs are "highly sensitive" to the a priori, even though their correlations in Table S3 are very similar (0.21 & 0.50 compared to 0.19 & 0.55). Also, line 492 calls the a priori profiles the "main cause" of the differences. From the information given, AMFs seem equally sensitive to both a priori and albedo, however the text suggests otherwise and should be rephrased.

Line 490: It's not clear how the use of an averaging kernel will reduce the effect of the a priori. Averaging kernels are most frequently used for making comparisons between two models, or between a model and a retrieved observation, in order to reduce errors that may arise when two models are based on different a priori profiles. From my understanding this does not reduce the retrieval's sensitivity to the a priori itself. This statement needs further explanation.

Technical notes:

Line 43: "20-50% from typical VCDs uncertainties of 40-60%" is ambiguous. Is it 20-50% of the typical uncertainty, or is it 20-50 percentage points. Consider rewording.

Figure 4: The green and red lines are hard to distinguish from the others in parts b-f. I realize that this is because they are overlapping, but (for example) in 4b are the green and red lines under the blue one, or under the pink one? It would be good to find a way to make this clearer.

Line 426: "...when cloud fraction is less than 0.1..." Should this be "greater than 0.1"? "Less than" seems to contradict the discussion in Section 3.3.2.

Lines 390-405/Figure 7: The text discusses relative differences in AMFs, and mentions that differences are small in unpolluted situations with larger differences in polluted areas. Figure 7 shows absolute differences in AMF, and the polluted and unpolluted plots have a similar vertical range. I would suggest including a plot of the relative differences to better illustrate the conclusions made in the text.

---

## Author Comment (AC1) · 19 Jan 2017

Referee (#2)

The authors would like to thank Referee #2 for his/her thoughtful and helpful comments and suggestions. Below are the comments by Referee #2 in blue and answers in black. Any modification made to the text has been highlighted within a green box. The line numbers correspond to the version of the manuscript available for online discussion.

**Comment (1):** Since the structural error is just the 1-sigma variation among the retrievals, I wonder if the fact that so many of them use the Kleipool albedo and fairly coarse models is underestimating the uncertainty due to albedo and profile shape. Thorne et al. (2005) specified that structural uncertainty should be "aggregated over many independent, plausibly constructed datasets..." The paper should address whether the uncertainty due to parameters shared among a large percentage of the retrievals used in the study may bias the interpretation to an underestimate of the uncertainty.

The principal point raised by the reviewer is a good one. However, only 3 out of 7 groups are using the exact same albedo values in their retrievals. It could still be argued that most of the albedo datasets (5 out of 7) come from the Kleipool et al. (2008) database. Therefore the estimation of the structural uncertainty could indeed be biased by the surface albedo originating from the same database (though represented differently) for most of the retrieval groups.

In order to test this hypothesis, we have estimated the structural uncertainty using 3 groups that use distinctly different surface albedo values (BIRA-IASB (MODIS BSA + OMI min LER for ocean and gap filling), KNMI/WUR (minimum LER from Kleipool et al. (2008)) and University of Leicester (mode LER from Kleipool et al. (2008)) and two groups that use exactly the same albedo (KNMI/WUR and NASA, both minimum LER from Kleipool et al. (2008)). The table below shows the structural uncertainty estimated for these two different ensembles and the 1-sigma relative uncertainty of the albedo values from the different datasets:

|  | Institutes | 1sigma of albedo datasets | Structural AMF uncertainty |
|---|---|---|---|
| **Identical albedo datasets** | KNMI/WUR, NASA | 0 | 19.7% |
| **Different albedo datasets** | BIRA-IASB, KNMI/WUR, U. Leicester | 30% | 17.7% |

We conclude that the estimation of the structural uncertainty is of the same order for the two different retrieval ensembles, so the fact that the surface albedos values come from the same database does not appear to be a clear driver of the overall structural uncertainty calculation. Nevertheless, the structural uncertainty using only two or three retrievals is smaller than the overall structural uncertainty calculated with the 7 different groups. This indicates that two retrievals only are insufficient to represent all the structural differences (use of BRDF for the surface reflectivity, different surface pressure values, aerosol corrections, cloud corrections,...) that are represented by the ensemble of 7 retrieval groups used in the manuscript.

We have added some discussion on this topic in line 436 (p. 19):

"Most of the surface albedo values used in the retrievals come from the Kleipool et al. (2008) database, which is based on OMI surface reflectance climatology. However, due to the different representations of surface reflectance within this database, only three retrieval groups use the exact same albedo values. We investigated if this could bias the estimation of the AMF structural uncertainty, and we concluded that that is not a clear driver of the overall structural uncertainty calculation."

**Comment (2):** Regarding the discussion in the final paragraph of p. 21, the point that validation of the a priori profiles is important is well taken, and I agree that estimating the effect of only the spatial (or temporal) resolution of the chemical transport model on the retrieval would require a very specific study. However our understanding of structural uncertainty is that comparing the AMFs calculated by a variety of retrievals allows a characterization of the total error independent of the parametric uncertainty calculations.

Given that the highest resolution a priori profiles used here were the 0.5 x 0.667 degree profiles in the POMINO retrieval, and that Valin et al. (2011), Heckel et al. (2011), and Yamaji et al (2014) all show that model resolution < 20 km is necessary to capture the nonlinearity of NOx chemistry; that Russell et al. (2011), McLinden et al. (2014), and Kuhlmann et al. (2015) show profiles at <= 15 km resolution significantly change the AMF, and Vinken et al. (2014) used a sub-grid plume parameterization in their retrieval with 0.5 x 0.667 degree profile resolution to a similar effect, my concern is that the overall uncertainty in the AMF derived from a structural uncertainty that does not include any retrievals using profiles with < 20 km resolution misrepresents the true uncertainty and bias. If adding at least one retrieval with < 20 km resolution a priori profile is impractical at this point, then at a minimum an extended discussion of the likelihood that the AMF uncertainty derived here is underestimated should be developed.

Unfortunately there is not yet a global retrieval that uses high-resolution a priori $NO_2$ profiles on a global scale; the specific retrievals are available only for particular regions (such as at city or regional scale, oil sands, shipping lanes) and particular studies. Ideally one would have to create a global AMF dataset using high resolution a priori profiles. This was not the main goal of this study and because of time constraints we will add some discussion on the topic, which we believe is very relevant both for this study and for satellite retrievals for current and future missions.

The table below shows the main quantitative findings from different studies (Kuhlmann et al. (2015), Mclinden et al. (2014), Heckel et al. (2011)) on the effect that using a high resolution a priori $NO_2$ profile have on specifically on AMF values:

| Study | AMF coarse | AMF high-res | Notes |
|---|---|---|---|
| **Kuhlmann et al. (2015)** | 1.19 GEOS-Chem | 0.82 CMAQ | Profiles not validated 40% smaller $AMF_{HR}$ |
| **McLinden et al. (2014)** | AMF coarse | AMF coarse / 1.9 | DOMINO:EC AMF ratio = 1.9 around emission sources LUT have very few points |
| **Heckel et al. (2011)** | = | = | 50% underestimation over land with coarse CTMs. |

Based on this simple literature survey, we created different high-resolution AMF databases in which the AMFs are 50% smaller over polluted areas to simulate the effect that using a high resolution a priori profile has on the AMFs. To create these simulated high-resolution AMFs we applied a 50% reduction to one of the members that participated in the comparison. Then we included this "new" AMF member in the comparison and performed the complete analysis as done in the manuscript. In this way we obtain a new AMF structural uncertainty estimate. By including or excluding the "new" members we may obtain an estimate of how much our original structural uncertainty may be biased because of the lack of a high-resolution AMF dataset in the ensemble. We did the following three experiments:

Exp. 1: Applied 50% reduction to NASA AMFs to create the new database (HR1)
Exp. 2: Applied 50% reduction to U. of Leicester AMFs to create the new database (HR2)
Exp. 3: Applied 50% reduction to U. of Leicester AMFs and WUR AMFs to create two databases (HR31, HR32).

The statistics of the comparison for each of the experiments is summarized in the tables and figures below (in line with Table S6 and Fig. 9 in the manuscript):

| Experiment 1 | | | | | | |
|---|---|---|---|---|---|---|
| | FEBRUARY | | | AUGUST | | |
| | Mean | Median | σ | Mean | Median | σ |
| BIRA | -23 | -24 | 16 | -18 | -21 | 16 |
| IUP-UB | -5 | -5 | 21 | -15 | -14 | 16 |
| Leicester Uni. | -4 | -4 | 16 | -6 | -5 | 11 |
| MPIC | -16 | 0 | 43 | -5 | 3 | 34 |
| NASA | -9 | -9 | 11 | -9 | -9 | 11 |
| WUR | 12 | 12 | 14 | 9 | 7 | 11 |
| HR1 | 45 | 45 | 6 | 45 | 45 | 6 |

**Table 1**: Statistical parameters of the comparison with the model mean in experiment 1 ($((\overline{AMF} - AMF_x)/\overline{AMF})*100$, in %) of total tropospheric AMFs over the globe for polluted pixels ($>1 \cdot 10^{15}$ molec/cm$^2$)

[Figure]

**Figure 1**: Ratio of tropospheric NO$_2$ AMFs by each group to the ensemble mean (left axis, bars) and the correlation coefficient (right axis, cross) for experiment 1.

| | Experiment 2 | | | | | |
|---|---|---|---|---|---|---|
| | FEBRUARY | | | AUGUST | | |
| | Mean | Median | σ | Mean | Median | σ |
| **BIRA** | -23 | 25 | 16 | -18 | -21 | 17 |
| **IUP-UB** | -5 | -5 | 21 | -15 | -14 | 16 |
| **Leicester Uni.** | -4 | -4 | 14 | -6 | -5 | 10 |
| **MPIC** | -16 | -1 | 45 | -5 | 3 | 34 |
| **NASA** | -9 | -9 | 12 | -9 | -9 | 12 |
| **WUR** | 11 | 11 | 13 | 9 | 7 | 11 |
| **HR2** | 48 | 48 | 7 | 46 | 47 | 5 |

**Table 2**: As Table 1 but for experiment 2

[Figure]

**Figure 2**: As Fig. 1 for experiment 2.

| | Experiment 3 | | | | | |
|---|---|---|---|---|---|---|
| | FEBRUARY | | | AUGUST | | |
| | Mean | Median | σ | Mean | Median | σ |
| **BIRA** | -32 | -34 | 17 | -27 | -30 | 18 |
| **IUP-UB** | -13 | -13 | 23 | -24 | -22 | 18 |
| **Leicester Uni.** | -11 | -11 | 15 | -13 | -14 | 11 |
| **MPIC** | -25 | -7 | 42 | -13 | -3 | 37 |
| **NASA** | -17 | -17 | 13 | -17 | -17 | 13 |
| **WUR** | 5 | 5 | 13 | 3 | 1 | 11 |
| **HR2** | 44 | 44 | 8 | 43 | 43 | 6 |

**Table 3**: As Table 1 but for experiment 3

[Figure]

**Figure 3**: As Fig. 1 for experiment 3.

We compared the standard deviations in these tables to Table S6 in the supplement. For the individual comparisons, the standard deviations with respect to the model mean do not change considerably, the order of magnitude stays within a couple of percent points. In terms of the estimation of the AMF structural uncertainty we conclude that:

1. AMF structural uncertainty over polluted areas ($>1*10^{15}$ molec/cm$^2$) increases by 1% in February and 3% in August.

2. AMF structural uncertainty over polluted areas ($>1*10^{15}$ molec/cm$^2$) increases by 3% in February and 3% in August.

3. AMF structural uncertainty over polluted areas ($>1*10^{15}$ molec/cm$^2$) does not increase in February and increases 6% in August.

These results indicate that the effect of lacking a hi-res AMF member on our AMF structural uncertainty is likely not very strong. The effect of course is notable in the AMF values as showed in the mentioned studies and also visible in the different figures of the different experiments. However, the original ensemble of 7 retrievals used in the comparison accounts for most of the possible structural differences in the AMF calculation.

We have extended the discussion on the effect of the a priori profiles in page 21: (two first paragraphs are already in the original manuscript, for context purpose)

Selecting a specific chemistry transport model thus influences the AMF structural uncertainty via differences in the profile shape. These differences in the profile shape depend on the different characteristics of the models (e.g. spatial and temporal resolution and parameterization of different processes in the atmosphere). Previous studies analysed how using different CTMs influences the $NO_2$ retrievals due to the change in the profile shapes used to calculate the AMF values. Heckel et al. (2011) compared retrievals using fine and coarse resolution models and concluded that using one AMF value for a large heterogeneous scene can lead to 50% bias in the retrieved $NO_2$ columns. Vinken et al. (2014) reported much smaller average differences of 10% in retrieved $NO_2$ columns mainly due to different emission inventories used in TM4 (3° x 2°) and GEOS-Chem (0.5° x 0.67°). According to Laughner et al. (2016), different temporal resolution also influences a priori profile shapes; they found differences in the retrieved $NO_2$ column for individual days up to 40% that were mostly explained by day-to-day wind direction variations that were not captured in the monthly averages.

All these aspects influence the estimation of retrieval (and AMF) theoretical uncertainties. In order to quantitatively estimate the effect of one model characteristic alone (e.g. the spatial resolution) on the AMF structural uncertainty it would be necessary to compare AMF calculated with the same approach but with just that specific characteristic being different in the profile shapes generated by the CTM. Such a specific sensitivity analysis has not been done in this study but should be considered in future AMF comparisons.

*To test the robustness of our structural uncertainty estimate, we did some experiments by creating new AMF databases to simulate the effect of high resolution a priori profiles on AMF values. Kulhmann et al. (2015), McLinden et al. (2014) and Heckel et*

*al. (2011) reported that AMFs calculated using coarse resolution a priori profiles are overestimated over polluted areas by approximately 50%. When including synthetic AMF databases emulating the use of high resolution a priori profiles, the estimated AMF structural uncertainty is not strongly affected (increases by 3-6%). This indicates that with the ensemble of retrievals used in our comparison the estimate of the structural uncertainty in the AMF calculation may be considered a robust estimate.*

**Comment (3):** The final statement on p. 21: "It is worth to note that using averaging kernels will reduce the effect of the a priori trace gas profile chosen in the retrieval scheme." Requires additional discussion. My understanding of the use of averaging kernels is that they are useful in two ways:
1) When comparing satellite retrieved VCDs against a model, applying the AKs to the model effectively "retrieves" the model trace gas profile, thus the dependence on the a priori profile in the retrieval is the same for both the observed and modeled column, and cancels out (Eskes and Boersma, 2003).
2) Alternately, one could use AKs to implement one's own a priori profiles in the retrieval.
Only in the first case would I say that the dependence on the trace gas profile is reduced, and that only applies when comparing to a model. Work using the satellite columns directly (e.g. Duncan et al. 2010, Beirle et al. 2011, Valin et al. 2013, Mebust and Cohen 2014, Lu et al. 2015, Liu et al. 2016, etc.) would not be able to use AKs in this way.

When averaging kernels are being applied, for instance when comparing retrieved $NO_2$ columns with modelled $NO_2$ distributions or with observed $NO_2$ profiles (aircraft, balloon), the comparison will become self-consistent in terms of using a priori information. Using the averaging kernel reduces systematic and random differences between modelled and satellite-observed columns because the representativeness of the modelled state for the observed state improves (e.g. Boersma et al., 2016). We agree that the retrieval of $NO_2$ columns will stay sensitive to the choice of the a priori profile, but using the averaging kernel provides a data user with the means to improve the consistency associated with the a priori profiles in interpreting the satellite data.

As both reviewers have raised their concern in this particular statement, we have tried to make it clearer:

It is worth to note that using averaging kernels in satellite applications (e.g. when comparing retrieved $NO_2$ columns with modelled $NO_2$ distributions or observed $NO_2$ profiles) will reduce the representativeness errors in the comparisons associated with the a priori trace gas profile used in the retrieval scheme (e.g. Boersma et al., 2016).

**Technical corrections:**
p. 21 l. 476 - PRevious (the R should be lowercase) - corrected
p. 21 l. 479 - NO2 (the 2 should be subscript) - corrected
p. 21 l. 480 - (Laughner et al. 2016) > Laughner et al. (2016) - modified

**References**

Beirle et al.: Megacity Emissions and Lifetimes of Nitrogen Oxides Probed from Space, Science, 333, 1737-1739, 2011.

Duncan et al.: Application of OMI observations to a space-based indicator of $NO_x$ and VOC controls on surface ozone formation, Atmos. Environ., 44, 2213-2223, doi: 10.1016/j.atmosenv.2010.03.010, 2010.

Eskes, H. J. and Boersma, K. F.: Averaging kernels for DOAS total-column satellite retrievals, Atmos. Chem. Phys., 3, 1285-1291, doi:10.5194/acp-3-1285-2003, 2003.

Heckel et al.: Influence of low spatial resolution a priori data on tropospheric $NO_2$ satellite retrievals, Atmos. Meas. Tech., 4, 1805-1820, doi: 10.5194/amt-4-1805-2011, 2011.

Kleipool, et al.: Earth surface reflectance climatology from 3 years of OMI data, J. Geophys. Res.: Atmospheres, 113, doi:10.1029/2008JD010290, d18308, 2008.

Kuhlmann et al.: Development of a custom OMI $NO_2$ data product for evaluating biases in a regional chemistry transport model, Atmos. Chem. Phys., 15, 5627-5644, doi:10.5194/acp-15-5627-2015, 2015.

Laughner, J. L., Zare, A., and Cohen, R. C.: Effects of daily meteorology on the interpretation of space-based remote sensing of $NO_2$, Atmos. Chem. Phys. Discuss., 2016, 1–27, doi:10.5194/acp-2016-536, 2016.

Liu et al.: $NO_x$ lifetimes and emissions of cities and power plants in polluted background estimated by satellite observations, Atmos. Chem. Phys., 16, 5283–5298, doi:10.5194/acp-16-5283-2016, 2016.

Lu et al., Emissions of nitrogen oxides from US urban areas: estimation from Ozone Monitoring Instrument retrievals for 2005–2014, Atmos. Chem. Phys., 15, 10367-10383, doi: 10.5194/acp-15-10367-2015, 2015.

McLinden et al., Improved satellite retrievals of $NO_2$ and $SO_2$ over the Canadian oil sands and comparisons with surface measurements, Atmos. Chem. Phys., 14, 3637-3656,doi:10.5194/acp-14-3637-2014, 2014

Mebust and Cohen, Space-based observations of fire NOx emissions coefficients: a global biome-scale comparison, Atmos. Chem. Phys., 14, 2509-2524, doi:10.5194/acp-14-2509-2014, 2014.

Russell et al., A high spatial resolution retrieval of $NO_2$ columns densities from OMI: method and evaluation, Atmos. Chem. Phys., 11, 8543-8554, doi:10.5194/acp-11-8543-2011, 2011.

Thorne, P. W., Parker, D. E., Christy, J. R., and Mears, C. A.: Uncertainties in climate trends: Lessons from Upper-Air Temperature Records, Bull. Am. Meteorol. Soc., 86, 1437–1442, doi:10.1175/BAMS-86-10-1437, 2005.

Valin et al., Effects of model resolution on the interpretation of satellite $NO_2$ observations, Atmos. Chem. Phys., 11, 11647-11655, doi:10.5194/acp-11-11647-2011, 2011.

Valin et al., Variations of OH radical in an urban plume inferred from $NO_2$ column measurements, Geophys. Res. Lett., 40, 1856–1860, doi:10.1002/grl.50267, 2013.

Vinken et al., Constraints on ship $NO_x$ emissions in Europe using GEOS-Chem and OMI satellite $NO_2$ observations, Atmos. Chem. Phys., 14, 1353–1369, 2014.

Yamaji et al., Influence of model-grid resolution on $NO_2$ vertical column densities over East Asia, J. Air. Waste. Manage., 64, 436-444, doi:10.1080/10962247.2013.827603, 2014.

---

## Author Comment (AC2) · 19 Jan 2017

Referee (#1)

The authors would like to thank Referee #1 for his/her thoughtful and helpful comments and suggestions. Below are the comments by Referee #1 in blue and answers in black. Any modification made to the text has been highlighted within a green box. The line numbers correspond to the version of the manuscript available for online discussion.

Line 132: "It is desirable to use as much information as possible retrieved from the satellite instrument itself." I didn't see much in the model descriptions with regards to this statement. What information from the satellite is used, is it the same for all the models, etc.

**Answer:** In the AMF calculation, usually as much information as possible is used from the satellite itself (in this case measurements from OMI). These are the viewing geometry, surface albedo (from the OMI LER climatology), cloud fraction and cloud pressure (from the OMI O2-O2 algorithm). The external parameters are terrain height (from a database) and a priori profile shape (from chemistry transport model simulations). This is the case for most of the retrievals that participated in the comparison. The sources for the different parameters in each of the retrievals are summarized in Table 3.

We have slightly modified the sentence:

"It is desirable to use as many forward model parameters as possible retrieved from the satellite instrument itself."

**Line 275**: "The agreement in this study is better than previous RTM comparisons:" Any idea why?

**Answer:** In this study we make a very detailed comparison of TOA reflectances in order to have a quantitative and specific number for the agreement between the RTMs. The comparison by Stammes (2001) only included two RTMs (DAK and MODTRAN), and Wagner et al. (2007) did not go as much in detail as we do here to get a TOA reflectance structural uncertainty value. Both studies give a more general value for TOA reflectance differences "up to or within 5%". Another general reason might be that RTMs have improved over the last 10-15 years.

Line 357 (or 337): The differences between online and LUT AMFs would depend on the resolution of the LUT. Since the Castellanos et al (2015) study indicated an 8% error (which is larger than the error due to RTM treatments you found), it would be worthwhile to have a brief discussion on the LUTs used in the different models and possible differences that may arise.

**Answer**: In Sect. 3.3.1 we obtain differences in AMF (using the same a priori information) of 6.5% for polluted areas and 2.5% for unpolluted areas. These numbers represent the differences introduced in the AMF calculation by (1) the use of different RTMs and (2) interpolation errors due to the use of LUTs. Other studies like Castellanos et al. (2015) found an average difference of 1% (and less than 8% for individual measurements) between interpolated LUT and online AMFs. Lin et al. (2014) found 1 to 5% differences in VCDs retrieved with interpolated LUTs and with online radiative transfer over China. These numbers are of the same magnitude as the

differences we found in our comparison. Furthermore, in our study interpolation errors are likely much less of a concern than in Castellanos et al. (2015) and Lin et al. (2014) because of the coarse grid used in their LUTs (this of course does not make those errors go away e.g. in DOMINO-2).

Based on this we conclude that the differences that may arise from using different LUTs by the different retrievals are not larger than 6.5%. Following the suggestion from the reviewer we add some discussion on this topic:

**L376**: "6.5% represents an upper limit value for the differences that using different RMTs and LUTs may introduce in the final AMF calculation."

**L436**: "Different groups use different LUTs for their AMF calculations, and POMINO uses pixel-by-pixel online radiative transfer calculations. The LUTs are different in several aspects: the RTMs used to create them and the number of reference points for each dimension. All these differences affect the AMF structural uncertainty. Based on the discussion in previous sections we consider that the use of different LUTs introduces a structural uncertainty of the order of 6.5%."

Section 3.3.2: Which model is used to evaluate the cloud corrections?

**Answer:** The AMFs used to evaluate the cloud corrections in Sect. 3.3.2 are those from KNMI/WUR. We add a sentence to clarify this on the text (**line 390**):

"To quantify the differences between the two approaches, we compare here tropospheric  $NO_2$  AMFs calculated by WUR (see Table 3) with the IPA and CM approach for two complete days of OMI measurements (02 February 2005 and 16 August 2005)"

**Line 467-470**: Characterization of the sensitivity of AMF to albedo and to a priori profiles is inconsistent. Surface albedo is said to "explain some" of the difference, while the AMFs are "highly sensitive" to the a priori, even though their correlations in Table S3 are very similar (0.21 & 0.50 compared to 0.19 & 0.55). Also, line 492 calls the a priori profiles the "main cause" of the differences. From the information given, AMFs seem equally sensitive to both a priori and albedo, however the text suggests otherwise and should be rephrased.

**Answer:** Our argumentation is based on the fact that in our ensemble, the number of pixels where albedo differences ( $\Delta A_s$ ) co-vary with AMF differences ( $\Delta AMF$ ) is considerably smaller than the number of pixels where modelled NO2 vertical column differences ( $\Delta NO_2$ ) co-vary with AMF differences (5382 vs. 15142 in 16 August and 1876 vs. 6483 in 02 February).

However, we agree with the reviewer that the albedo differences are important for the AMF differences and that the original text might suggest that the influence of surface albedo is not really important. Therefore, we have rephrased the sentences in lines 467-473 to make it clearer.

"...and that surface albedo differences explain WUR and BIRA AMF differences especially in winter, when  $NO_2$  is found close to the surface..."

"In our ensemble, the WUR-BIRA AMF differences are highly sensitive to the differences between the a priori  $NO_2$  profiles used, especially in summer."

**Line 492**: "In the previous section, we found that differences between a priori  $NO_2$  profiles and the surface albedo are the main cause for AMF structural uncertainty when cloud parameters are identical."

Line 490: It's not clear how the use of an averaging kernel will reduce the effect of the a priori. Averaging kernels are most frequently used for making comparisons between two models, or between a model and a retrieved observation, in order to reduce errors that may arise when two models are based on different a priori profiles. From my understanding this does not reduce the retrieval's sensitivity to the a priori itself. This statement needs further explanation.

When averaging kernels are being applied, for instance when comparing retrieved  $NO_2$  columns with modelled  $NO_2$  distributions or with observed  $NO_2$  profiles (aircraft, balloon), the comparison will become self-consistent in terms of using a priori information. Using the averaging kernel reduces systematic and random differences between modelled and satellite-observed columns because the representativeness of the modelled state for the observed state improves (e.g. Boersma et al., 2016). We agree that the retrieval of  $NO_2$  columns will stay sensitive to the choice of the a priori profile, but using the averaging kernel provides a data user with the means to improve the consistency associated with the a priori profiles in interpreting the satellite data.

As both reviewers have raised their concern in this particular statement, we have tried to make it clearer:

It is worth to note that using averaging kernels in satellite applications (e.g. when comparing retrieved NO2 columns with modelled NO2 distributions or observed NO2 profiles) will reduce the representativeness errors in the comparisons associated with the a priori trace gas profile used in the retrieval scheme (e.g. Boersma et al., 2016).

**Technical notes:**

**Line 43**: "20-50% from typical VCDs uncertainties of 40-60%" is ambiguous. Is it 20-50% of the typical uncertainty, or is it 20-50 percentage points. Consider rewording.

**Answer:** We have now rephrased it:**

Previous studies indicated that AMF calculation is the largest source of uncertainty (contributing up to half of the typical VCD uncertainties of 40-60%.) in the  $NO_2$  and HCHO retrievals..."

**Figure 4**: The green and red lines are hard to distinguish from the others in parts b-f. I realize that this is because they are overlapping, but (for example) in 4b are the green and red lines under the blue one, or under the pink one? It would be good to find a way to make this clearer.

**Answer:** We have changed the figure, particularly we have now applied a different marker for each RTM. For example in Fig. 4(b) even if the pink line is still in front, by looking at the markers we can see that green and red lines are under the blue line and not under the pink line.

**Line 426**: "when cloud fraction is less than 0.1..." Should this be "greater than 0.1"? "Less than" seems to contradict the discussion in Section 3.3.2.

**Answer:** Yes, indeed. IUP-UB applied IPA cloud correction when cloud fraction is greater than 0.1. We have modified the text:

**"IUP-UB and BIRA now apply IPA only when cloud fraction exceeds 0.1 and 0.2 respectively"**

**Lines 390-405/Figure 7**: The text discusses relative differences in AMFs, and mentions that differences are small in unpolluted situations with larger differences in polluted areas. Figure 7 shows absolute differences in AMF, and the polluted and unpolluted plots have a similar vertical range. I would suggest including a plot of the relative differences to better illustrate the conclusions made in the text.

**Answer:** Thanks for this comment. Following the reviewer's suggestion, we have substituted Fig. 7 with the relative differences in the y-axis (see figure below).

**References**

Boersma, K. F., Vinken, G. C. M., and Eskes, H. J.: Representativeness errors in comparing chemistry transport and chemistry climate models with satellite UV-Vis tropospheric column retrievals, Geosci. Mod. Dev., 9, 875–898, doi:10.5194/gmd-9-875-2016, 2016.

Castellanos, P., Boersma, K. F., Torres, O., and de Haan, J. F.: OMI tropospheric NO2 air mass factors over South America: effects of biomass burning aerosols, Atmos. Meas. Tech., 8, 3831–3849, doi:10.5194/amt-8-3831-2015, 2015.

Lin, J.-T., Martin, R. V., Boersma, K. F., Sneep, M., Stammes, P., Spurr, R., Wang, P., Van Roozendael, M., Clémer, K., and Irie, H.: Retrieving tropospheric nitrogen dioxide from the Ozone Monitoring Instrument: effects of aerosols, surface reflectance anisotropy, and 725 vertical profile of nitrogen dioxide, Atmos. Chem. Phys., 14, 1441–1461, doi:10.5194/acp-14-1441-2014, 2014.

Stammes, P.: Spectral radiance modeling in the UV-Visible range, IRS2000: in: Current problems in atmospheric radiation, edited by: Smith, W. L. and Timofeyev, Y. J., pp. 385–388, 2001.

Wagner, T., Burrows, J. P., Deutschmann, T., Dix, B., von Friedeburg, C., Frieß, U., Hendrick, F., Heue, K.-P., Irie, H., Iwabuchi, H., Kanaya, Y., Keller, J., McLinden, C. A., Oetjen, H., Palazzi, E., Petritoli, A., Platt, U., Postylyakov, O., Pukite, J., Richter, A., van Roozendael, M., Rozanov, A., Rozanov, V., Sinreich, R., Sanghavi, S., and Wittrock, F.: Comparison of box-air-mass-factors and radiances for Multiple-Axis Differential Optical Absorption Spectroscopy (MAX-DOAS) geometries calculated from different UV/visible radiative transfer models, Atmos. Chem. Phys., 7, 1809– 1833, doi:10.5194/acp-7-1809-2007, 2007.